# Structure of the dopamine $D_2$ receptor in complex with the antipsychotic drug spiperone

Dohyun Im[1], Asuka Inoue[2,3,4], Takaaki Fujiwara[1,9], Takanori Nakane [5,10], Yasuaki Yamanaka[1], Tomoko Uemura[1], Chihiro Mori[1], Yuki Shiimura [1,6], Kanako Terakado Kimura[1], Hidetsugu Asada [1], Norimichi Nomura [1], Tomoyuki Tanaka[1,7], Ayumi Yamashita[1,7], Eriko Nango[7,9], Kensuke Tono[8], Francois Marie Ngako Kadji [2], Junken Aoki [2,4,11], So Iwata [1,7✉] & Tatsuro Shimamura [1✉]

In addition to the serotonin 5-HT$_{2A}$ receptor (5-HT$_{2A}$R), the dopamine D$_2$ receptor (D$_2$R) is a key therapeutic target of antipsychotics for the treatment of schizophrenia. The inactive state structures of D$_2$R have been described in complex with the inverse agonists risperidone (D$_2$R$_{ris}$) and haloperidol (D$_2$R$_{hal}$). Here we describe the structure of human D$_2$R in complex with spiperone (D$_2$R$_{spi}$). In D$_2$R$_{spi}$, the conformation of the extracellular loop (ECL) 2, which composes the ligand-binding pocket, was substantially different from those in D$_2$R$_{ris}$ and D$_2$R$_{hal}$, demonstrating that ECL2 in D$_2$R is highly dynamic. Moreover, D$_2$R$_{spi}$ exhibited an extended binding pocket to accommodate spiperone's phenyl ring, which probably contributes to the selectivity of spiperone to D$_2$R and 5-HT$_{2A}$R. Together with D$_2$R$_{ris}$ and D$_2$R$_{hal}$, the structural information of D$_2$R$_{spi}$ should be of value for designing novel antipsychotics with improved safety and efficacy.

[1] Department of Cell Biology, Graduate School of Medicine, Kyoto University, Kyoto, Japan. [2] Graduate School of Pharmaceutical Sciences, Tohoku University, Sendai, Miyagi, Japan. [3] Advanced Research & Development Programs for Medical Innovation (PRIME), Japan Agency for Medical Research and Development (AMED), Chiyoda, Tokyo, Japan. [4] Advanced Research & Development Programs for Medical Innovation (LEAP), AMED, Chiyoda, Tokyo, Japan. [5] Department of Biological Sciences, Graduate School of Science, University of Tokyo, Bunkyo, Tokyo, Japan. [6] Molecular Genetics, Institute of Life Science, Kurume University, Kurume, Fukuoka, Japan. [7] RIKEN SPring-8 Center, Sayo, Hyogo, Japan. [8] Japan Synchrotron Radiation Research Institute, Sayo, Hyogo, Japan. [9] Present address: Institute of Multidisciplinary Research for Advanced Materials, Tohoku University, Sendai, Japan. [10] Present address: MRC Laboratory of Molecular Biology, Cambridge, UK. [11] Present address: Graduate School of Pharmaceutical Sciences, University of Tokyo, Bunkyo, Tokyo, Japan. ✉email: s.iwata@mfour.med.kyoto-u.ac.jp; t.shimamura@mfour.med.kyoto-u.ac.jp

Dopamine is a neurotransmitter that controls numerous physiologic functions in the brain and peripheral nervous system via dopamine receptors of the G-protein-coupled receptor (GPCR) superfamily. In humans, five dopamine receptors ($D_1R–D_5R$) have been identified and have been classified according to their sequence, intracellular signaling, pharmacology, and localization as $D_1$-class receptors ($D_1R$ and $D_5R$) or $D_2$-class receptors ($D_2R$, $D_3R$, and $D_4R$)[1–3]. Thus, $D_2R$ is similar to both $D_3R$ and $D_4R$, with 80 and 54% sequence identities, respectively, in their transmembrane helices[4–6]. $D_2R$ is highly distributed in the striatum, nucleus accumbens, and olfactory tubercle[7,8], and it plays important pharmacologic roles in numerous human disorders related to dopaminergic dysfunction, including schizophrenia[9–11] and Parkinson's disease[12,13].

$D_2R$ antagonists have been developed as antipsychotics to block dopaminergic transmission for the treatment of schizophrenia[14]. Antipsychotics are either typical or atypical; typical antipsychotics generally antagonize $D_2R$, whereas atypical antipsychotics antagonize both $D_2R$ and the serotonin $5-HT_{2A}$ receptor ($5-HT_{2A}R$). Both of these groups at least improve the positive symptoms of schizophrenia[15]. However, they also are associated with a wide range of severe side effects, such as extrapyramidal symptoms, weight gain, metabolic disorders, and constipation[16,17]. Extrapyramidal symptoms are caused by excessive inhibition of $D_2R$ in the nigrostriatal pathway. Other side effects are primarily due to the undesired binding of antipsychotics to other aminergic receptors, such as the serotonin $5-HT_{2C}$ receptor ($5-HT_{2C}R$), which exhibits 46% sequence identity with $5-HT_{2A}R$.

GPCR structures have been successfully utilized for the structure-guided discovery of new ligands[18,19]. In $D_2$-class receptors, $D_3R$ and $D_4R$ structures were determined in complex with the benzamide antipsychotics eticlopride and nemonapride ($D_3R_{eti}$ and $D_4R_{nem}$), respectively[20,21]. The inactive conformations of $D_2R$ have been described in complexes with risperidone ($D_2R_{ris}$), a pyridopyrimidine antipsychotic[22], and with haloperidol ($D_2R_{hal}$), a butyrophenone antipsychotic[23]. $D_2R_{ris}$ and $D_2R_{hal}$ are practically identical, as shown by the RMSD values of $C\alpha$ atoms between them (Supplementary Table 1). A G-protein-bound active conformation of $D_2R$ also was reported in complex with an agonist, bromocriptine ($D_2R_{bro}$)[24]. Interestingly, the conformation of the extracellular loop (ECL) 2 in $D_2R_{ris}$ and $D_2R_{hal}$ is entirely different from those of $D_3R_{eti}$, $D_4R_{nem}$, and $D_2R_{bro}$ (Supplementary Fig. 1). Additionally, while the conformation of ECL1 is relatively conserved among the structures of $D_2$-class receptors, Trp100[23.50] on ECL1 of $D_2R_{ris}$ is uniquely directed to the binding pocket (Supplementary Fig 1).

In this study, we describe the structure of $D_2R$ in complex with spiperone ($D_2R_{spi}$), a butyrophenone typical antipsychotic that binds with high affinity to $D_2R$, $D_3R$, $D_4R$, and $5-HT_{2A}R$[25]. We also present a structural comparison of $D_2R_{spi}$ with other $D_2R$ structures, $D_3R_{eti}$, and $D_4R_{nem}$, in addition to $5-HT_{2A}R$ complexed with risperidone ($5-HT_{2A}R_{ris}$)[26] and $5-HT_{2C}R$ with ritanserin ($5-HT_{2C}R_{rit}$)[27]. The structure of $D_2R_{spi}$ given herein provides valuable information for the rational design of antipsychotics with improved receptor selectivity.

## Results

### Overall structure of $D_2R_{spi}$.
Because wild-type $D_2R$ is not expressed in *Spodoptera frugiperda* (Sf9) insect cells, we prepared a stable construct for crystallization trials. $D_2R$ was stabilized by the truncation of 34 N-terminal residues and the replacement of the intracellular loop (ICL) 3 with the thermostabilized apocytochrome b562RIL[28] ($D_2R$-bRIL). $D_2R$-bRIL was further stabilized by the mutations S121K[3.39] and L123W[3.41] (here, superscripts

indicate residue numbers according to the Ballesteros–Weinstein scheme[15]) and the replacement of bRIL with mbIIG, the loop-modified cytochrome $b_{562}$IIG[29] ($D_2R$-mbIIG S121K[3.39]/L123W[3.41], see Methods). The use of mbIIG instead of bRIL was essential to obtain $D_2R$ crystals. S121K[3.39] is a mutation of the allosteric sodium ion binding site of class A GPCRs that mimics the presence of the sodium ion, therefore stabilizing the inactive state[30,31]. The S121K[3.39], L123W[3.41], and S121K[3.39]/L123W[3.41] mutants, in addition to the stabilized construct, showed similar affinities for spiperone to the wild-type human $D_2R$ (Supplementary Table 2 and Supplementary Fig. 2), suggesting that these mutations did not substantially affect the binding of spiperone. Additionally, the L123W[3.41] mutant showed similar antagonist activity to that of the wild-type human $D_2R$ by a transforming growth factor alpha (TGFα) shedding assay[32], which measured the antagonist activities of wild-type and mutant $D_2R$ for spiperone against a fixed concentration of dopamine (Supplementary Table 3 and Supplementary Fig. 3). However, because the S121K[3.39] mutation stabilizes the inactive state of $D_2R$, the antagonist activities of the mutants with S121K[3.39] could not be determined (Supplementary Table 3 and Supplementary Fig. 3). Eticlopride and sulpiride enhance the affinity for $D_2R$ in the presence of the sodium ion, whereas spiperone does not[33]. This enhancement is ascribed to an interaction network from the bound sodium ion in the allosteric binding site[33]. The S121K[3.39] mutation decreased the affinity for eticlopride (Supplementary Table 2), suggesting that the side chain of Lys121[3.39] does not sufficiently mimic the sodium ion in the allosteric binding site for the binding of eticlopride.

For crystallization, we generated an antibody recognizing the $D_2R$ structure (IgG3089) and prepared a novel Fab fragment (Fab3089) of IgG3089. Using the stabilized construct and Fab3089, we successfully obtained crystals of $D_2R$ (Supplementary Fig. 4a). The structure of $D_2R$ was determined in complex with spiperone at 3.1 Å resolution using an X-ray free-electron laser (Fig. 1, Table 1, and Supplementary Fig. 4). $D_2R_{spi}$ bound to Fab3089 at the extracellular region (Fig. 1a and Supplementary Fig. 4c), exhibiting a canonical GPCR fold with seven transmembrane helices (TM1–7) and an intracellular amphipathic helix 8 (H8) (Fig. 1). The $D_2R_{spi}$ structure demonstrated the inactive conformation because the seven helical bundle structure and the conformations of the four activation microswitches are more similar to those of the inactive state conformation in $D_2R_{ris}$ than those of the active state conformation in $D_2R_{bro}$ (Supplementary Fig. 5 and Supplementary Table 1). We also compared the PIF motif of $D_2R_{spi}$ with those of the inactive state and the active state structures of the $\beta_2$-adrenergic receptor to confirm $D_2R_{spi}$ demonstrated the inactive conformation, because $D_2R_{ris}$ contains the I122A[3.40] mutation in the motif.

There were no secondary structures in the ECLs and ICL1; ICL2 in $D_2R_{spi}$ was disordered. Spiperone was bound to the orthosteric binding site (Fig. 1 and Supplementary Fig. 4d). Like other class A GPCRs, the ligand-binding pocket was covered by the C-terminal segment of ECL2, which is stabilized by a disulfide bond between Cys107[3.25] on TM3 and Cys182[45.50] on ECL2[34] (Fig. 1).

**Binding mode of spiperone**. In $D_2R_{spi}$, spiperone was surrounded by residues from TM2, 3, 5, 6, and 7 and ECL2 (Figs. 1 and 2a, b). The binding mode of spiperone was consistent with the results of the TGFα shedding assay (Supplementary Table 3) and of the ligand-binding assay for spiperone using the mutants[35–38].

The tertiary amine in the triazaspiro ring formed a salt bridge with Asp114[3.32]; this is strictly conserved in aminergic receptor structures (Fig. 2a, b). In $D_2R$, the mutant of Asp114[3.32] loses its affinities to both agonists and antagonists[22,36]. This interaction

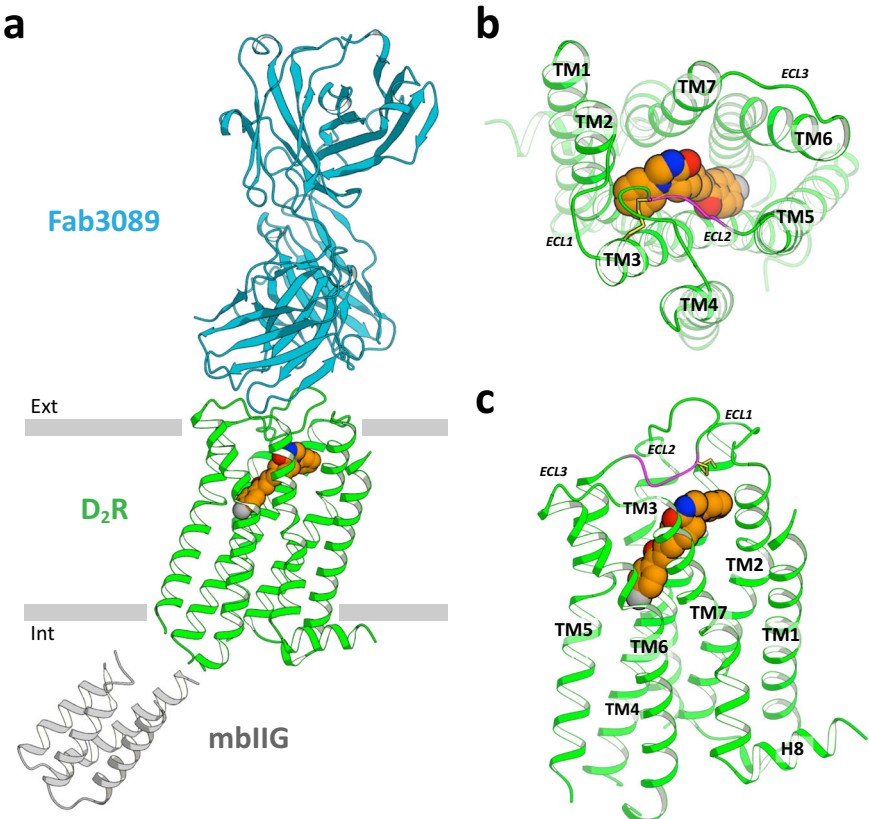

**Fig. 1 Structure of D$_2$R$_{spi}$ in complex with spiperone. a** Overall structure of D$_2$R$_{spi}$-Fab 3089 complex. Extracellular (**b**) and side (**c**) views of D$_2$R$_{spi}$. Spiperone, D$_2$R, Fab3089, and mbIIG are indicated in orange, green, cyan, and gray, respectively. The disulfide bond and the C-terminal segment of ECL2 are shown in yellow and magenta, respectively, in (**b**) and (**c**). Ext, extracellular; Int, intracellular.

| Table 1 Data collections and structure refinement statistics. | |
|---|---|
| | **D$_2$R$_{spi}$ (PDB 7DFP)** |
| **Data collection** | |
| Space group | C2 |
| Cell dimensions | |
| $a, b, c$ (Å) | 161.9, 40.5, 165.9 |
| $\alpha, \beta, \gamma$ (°) | 90, 116.5, 90 |
| Resolution (Å) | 43.1-3.1 (3.2-3.1)$^a$ |
| $R_{split}$ (%) | 18.8 (73.2) |
| CC$_{1/2}$ | 0.97 (0.58) |
| $I / \sigma(I)$ | 4.4 (1.5) |
| Completeness (%) | 100 (100) |
| Redundancy | 99.8 (45.4) |
| Refinement | |
| Resolution (Å) | 43.1-3.1 (3.2-3.1) |
| No. reflections | 18,048 |
| $R_{work} / R_{free}$ | 18.5 / 21.7 (26.7/31.7) |
| No. atoms | |
| Protein | 5896 |
| Ligand | 29 |
| B factors | |
| Protein | 97.1 |
| Ligand | 105.9 |
| R.M.S. deviations | |
| Bond lengths (Å) | 0.002 |
| Bond angles (°) | 0.56 |
| $^a$Values in parentheses are for highest-resolution shell. | |

may be stabilized by the conserved hydrogen bond between Asp114[3.32] and Tyr416[7.43] [39]. On the opposite side of spiperone from Asp114[3.32], there is hydrophobic contact between spiperone and Phe389[6.51]. This contact is likely essential for spiperone binding, given that F389A[6.51] showed a 34-fold reduction in affinity for spiperone compared to that of wild-type D$_2$R[35]. Additionally, there was contact between the triazaspiro ring and Ile183[45.51], Ile184[45.52], and Cys182[45.50] on ECL2 (Fig. 2a, b). The loss of this contact with the I184A[45.52] mutation significantly reduced the antagonist activity for spiperone (Supplementary Table 3), suggesting that the contact with Ile184[45.52] is crucial for the antagonist activity of spiperone. The contact between Ile183[45.51] and spiperone was influenced by the binding of Fab3089, because the side-chain conformation of Ile183[45.51] was stabilized by Tyr55 of Fab3089 (Fig. 2c). The I183A[45.51] mutant slightly increased the antagonist activity for spiperone (Supplementary Table 3).

On the extracellular side of the salt bridge, the phenyl ring of spiperone was bound in an extended binding pocket (EBP) between TM2 and TM3 (Fig. 2a, d). The EBP was formed by residues Val87[2.57], Trp90[2.60], Val91[2.61], Leu94[2.64], Trp100[23.50], Phe110[3.28], Val111[3.29], and Cys182[45.50]. The drastic reduction of the antagonist activity of W100A[23.50] for spiperone indicates that Trp100[23.50] is crucial for maintaining EBP conformation (Supplementary Table 3). Because of the triazaspiro ring rigidity and the direction of the conserved salt bridge between the tertiary amine and Asp114[3.32], the EBP is likely essential for spiperone binding. In the EBP of D$_2$R$_{spi}$, spiperone's phenyl ring forms hydrophobic contacts with Trp90[2.60], Val91[2.61], Leu94[2.64], Phe110[3.28], and Val111[3.29] (Fig. 2a, b). Mutations of most of

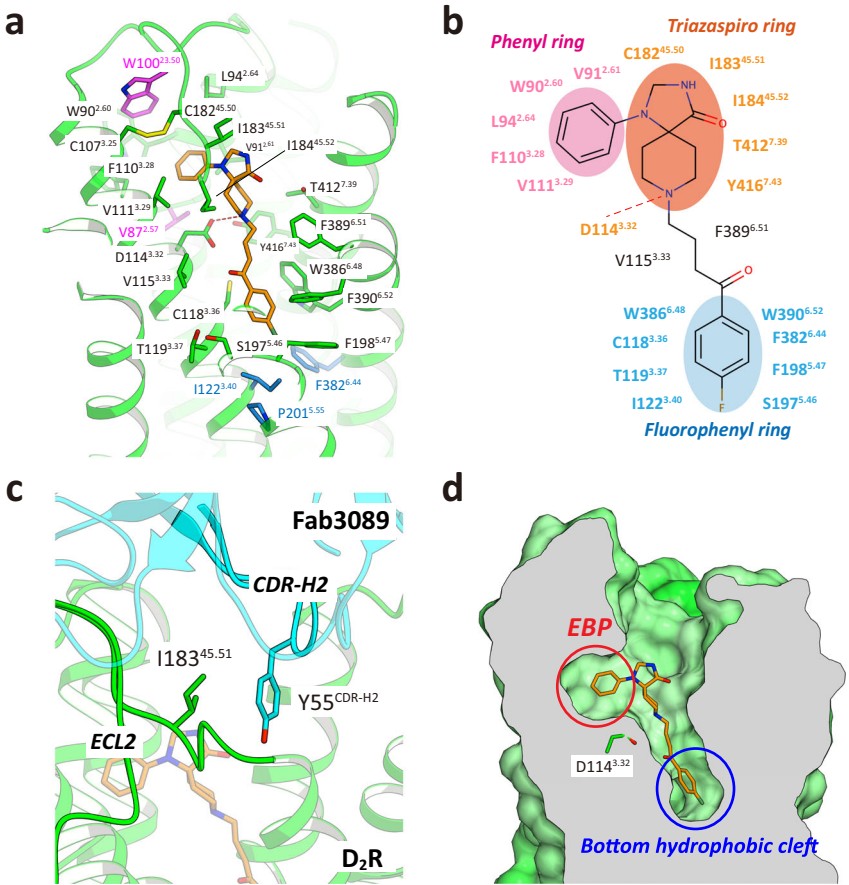

**Fig. 2 The ligand-binding pocket of $D_2R_{spi}$. a** Close-up view of the ligand-binding pocket of $D_2R_{spi}$. Spiperone and $D_2R$ are indicated in orange and green, respectively. The side chains of the contact residues within 4.5 Å of spiperone are shown as green sticks. The side chain of $W100^{23.50}$ is indicated in magenta. The side chains of the residues in the PIF motif are shown as blue sticks. **b** Diagram of the interactions between $D_2R$ and spiperone. **c** Fab3089 binding site. The side chains of $I183^{45.51}$ of $D_2R_{spi}$ and Y55 of Fab3089 are shown as green and cyan sticks, respectively. **d** Vertical cross section of (**a**). Red and blue circles indicate the EBP and the bottom hydrophobic cleft, respectively.

these residues resulted in considerably decreased antagonist activity for spiperone (Supplementary Table 3).

On the intracellular side of the salt bridge, spiperone's fluorophenyl ring penetrated deeply into the ligand-binding pocket, binding in the bottom hydrophobic cleft (Fig. 2a, b, d). Similar bottom hydrophobic clefts have been observed in the structures of the histamine $H_1$ receptor[40], $5-HT_{2A}R^{26}$, and $5-HT_{2C}R^{27}$. In the cleft, the fluorophenyl ring formed a CH–π interaction with $Cys118^{3.36}$; hydrophobic interactions with $Thr119^{3.37}$, $Ile122^{3.40}$, $Ser197^{5.46}$, $Phe198^{5.47}$, and $Phe382^{6.44}$; and edge-to-face π interactions with $Trp386^{6.48}$ and $Phe390^{6.52}$ (Fig. 2a, b). $Trp386^{6.48}$ is a microswitch in the CWxP motif. Indeed, $W386L^{6.48}$ affected activation by dopamine and showed no antagonist activity for spiperone (Supplementary Table 3 and Supplementary Fig. 3). The interaction with $Phe390^{6.52}$ is essential for the binding of spiperone, because $F390A^{6.52}$ drastically decreases the affinity for spiperone[35]. By contrast, contact with $Phe198^{5.47}$ is not essential for the binding of spiperone, given that $F198A^{5.47}$ showed a similar affinity for spiperone to that of wild-type $D_2R^{35}$. The $S197A^{5.46}$ mutant also showed an affinity for spiperone similar to that of the wild-type $D_2R^{36–38}$, thus strengthening contact between spiperone and the Cβ atom of $Ser197^{5.46}$.

$Ile122^{3.40}$ and $Phe382^{6.44}$ belong to the PIF motif located at the bottom of the ligand-binding pocket in aminergic receptors[41]. When activated, the PIF motif conformationally rearranges, with the outward movement of the cytoplasmic side of TM6. The PIF

motif conformation in $D_2R_{spi}$ is that of the inactive state (Supplementary Fig. 5). Thus, the direct interactions of spiperone with $Ile122^{3.40}$ (3.7 Å distance) and $Phe382^{6.44}$ (3.6 Å distance) in the PIF motif may block the conformational rearrangements of the PIF motif and help to stabilize the inactive conformation, as has been observed in the structural studies of $5-HT_{2A}R^{26}$ and $5-HT_{2C}R^{27}$. Of the 21 contact residues, 20 were conserved between $D_2R$ and $D_3R$ (Supplementary Table 4), reflecting the similarly high affinity of these receptors for spiperone[42].

**Comparison with $D_2R_{ris}$, $D_2R_{hal}$, and $D_2R_{bro}$.** There are striking structural differences in the ligand-binding pocket of $D_2R_{spi}$ and the other inactive state structures of $D_2R$: $D_2R_{ris}$ and $D_2R_{hal}$ (Fig. 3a, b and Supplementary Table 1). In $D_2R_{spi}$, the ligand-binding pocket was covered by the C-terminal segment of ECL2 (Fig. 3a–c), on which the side chains $Ile183^{45.51}$ and $Ile184^{45.52}$ pointed to the entrance and the bottom of the ligand-binding pocket, respectively. This conformation was conserved in the structures of other aminergic receptors, including $D_3R$, $D_4R$, and $5-HT_{2A}R$ (Supplementary Fig. 6). In $D_2R_{ris}$ and $D_2R_{hal}$, however, ECL2 extended away from the top of the receptor core (Fig. 3a, b, d). In this conformation, $Ile183^{45.51}$ was buried in the hydrophobic core outside the ligand-binding pocket, and $Ile184^{45.52}$ reached the top of the ligand-binding pocket (Fig. 3a, b). Thus, $Ile184^{45.52}$ did not contact with risperidone in $D_2R_{ris}$, while $Ile184^{45.52}$ contacted with spiperone in $D_2R_{spi}$. These findings

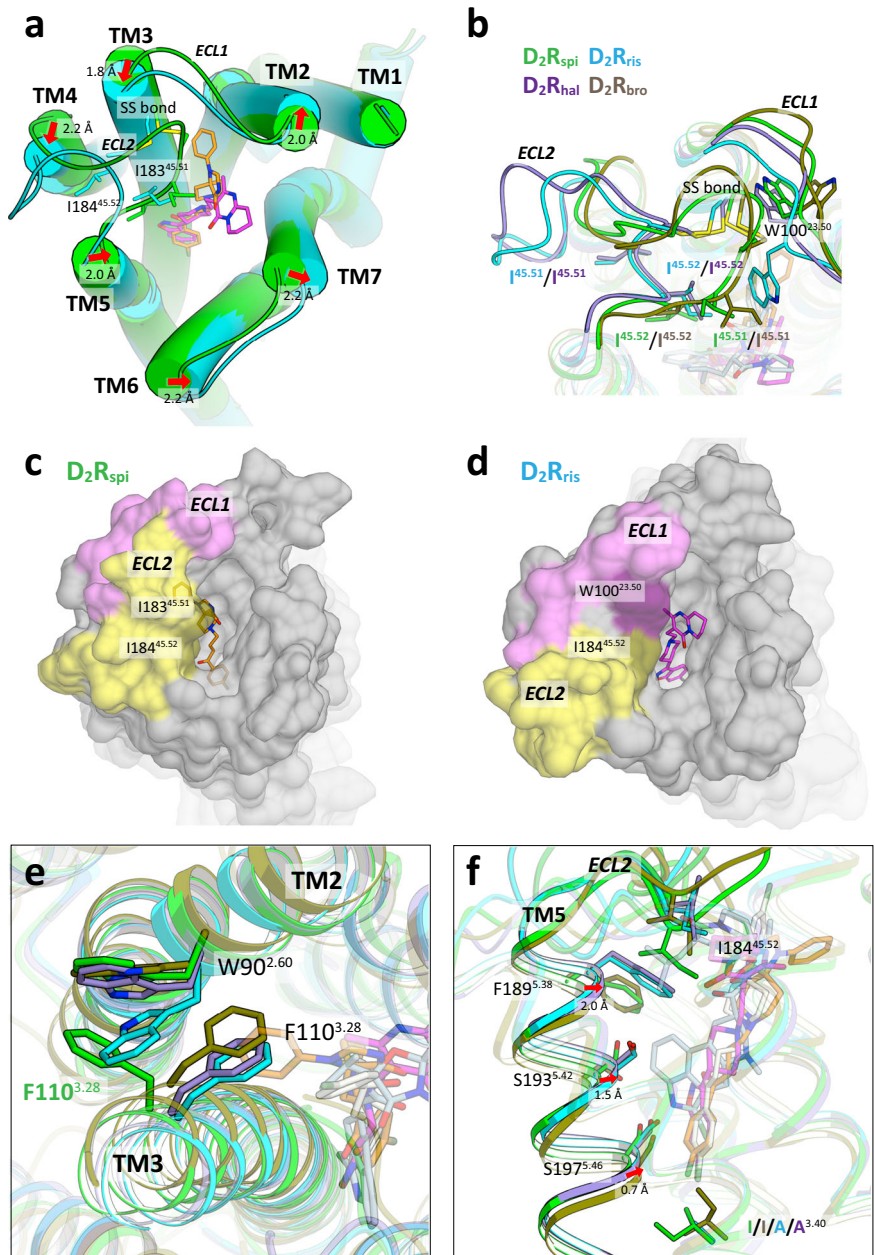

**Fig. 3 Comparison of D$_2$R structures. a** Extracellular view of the superposition of D$_2$R$_{spi}$ and D$_2$R$_{ris}$. The side chains of disulfide bridge, I183[45.51] and I184[45.52] are shown as sticks. Red arrows indicate the shift of helices in D$_2$R$_{ris}$ with the distance relative to D$_2$R$_{spi}$. **b** Extracellular view of ECL1 and ECL2 of D$_2$R$_{spi}$, D$_2$R$_{ris}$, D$_2$R$_{hal}$ and D$_2$R$_{bro}$. The side chains of the disulfide bridge, W100[23.50], I183[45.51], and I184[45.52] are shown as sticks. Surface representation of D$_2$R$_{spi}$ (**c**) and D$_2$R$_{ris}$ (**d**) viewed from the extracellular side. ECL1 and ECL2 are pink and yellow, respectively. **e** The EBP of D$_2$R$_{spi}$ and the corresponding part of D$_2$R$_{ris}$, D$_2$R$_{hal}$ and D$_2$R$_{bro}$. The side chains of W90[2.60] and F110[3.28] are shown as sticks. **f** Side view of the superposition of D$_2$R$_{spi}$, D$_2$R$_{ris}$, D$_2$R$_{hal}$ and D$_2$R$_{bro}$ around TM5 and ECL2. Red arrows indicate the shift of the extracellular half of TM5 in D$_2$R$_{ris}$ and D$_2$R$_{hal}$ relative to D$_2$R$_{spi}$. In (**b**), (**e**), and (**f**), D$_2$R$_{spi}$ (green), D$_2$R$_{ris}$ (cyan), D$_2$R$_{hal}$ (purple), D$_2$R$_{bro}$ (olive), spiperone (orange), risperidone (magenta), haloperidol (ivory), and bromocriptine (lightblue) are shown.

were consistent with I184A[45.52] showing similar antagonist activity for risperidone with the wild-type (Supplementary Table 3).

On ECL1, D$_2$R demonstrated the diverse side-chain conformation of Trp100[23.50]. In D$_2$R$_{spi}$, Trp100[23.50] interacted with the conserved disulfide bond (Fig. 3b). The conformation was highly conserved in the structures of class A GPCRs (Supplementary Fig. 6). The interaction between a disulfide bond and tryptophan is often observed in protein structures; this may contribute to protecting the disulfide bond and stabilizing the structure[43,44].

Trp100[23.50] in D$_2$R$_{hal}$ exhibited similar side-chain conformation with that of D$_2$R$_{spi}$, although it did not contact with the disulfide bond because of ELC2 flipping (Fig. 3b). In D$_2$R$_{ris}$, Trp100[23.50] moves to the ligand-binding pocket and forms a T-stacking interaction with risperidone's tetrahydropyridopyrimidinone ring[22] (Fig. 3b). Trp100[23.50] in D$_2$R$_{ris}$ forms a hydrophobic patch with Leu94[2.64] and Ile184[45.52], covering the ligand-binding pocket. Despite this, the ligand-binding pocket in D$_2$R$_{ris}$ was more exposed to the extracellular solution compared with that in D$_2$R$_{spi}$ (Fig. 3c, d). W100A[23.50] in D$_2$R has been shown to reduce

the residence times of several antipsychotics[22]. Based on these results, it was hypothesized that the hydrophobic patch in $D_2R_{ris}$ contributes to the slow dissociation of antipsychotics[22]. However, the results were also consistent with the conformation observed in $D_2R_{spi}$, in which Trp100[23.50] stabilized the conformation of ECL2 and EBP.

The EBP is uniquely observed in $D_2R_{spi}$ among the inactive state structures of $D_2R$. In $D_2R_{spi}$, the side chain of Phe110[3.28] that creates the EBP was flipped compared with those of $D_2R_{ris}$ and $D_2R_{hal}$ (Fig. 3e), $D_3R_{eti}$, and the 5-HT$_2$ receptors. In $D_2R_{spi}$, the flipped Phe110[3.28] side chain formed a stacking interaction with Trp90[2.60] (Fig. 3e).

$D_2R_{ris}$ and $D_2R_{hal}$ also possessed the bottom hydrophobic cleft (Fig. 3f). However, the conformation of this cleft in $D_2R_{ris}$ and $D_2R_{hal}$ was altered by the shift of the extracellular half of TM5 in the ligand-binding pocket relative to that in $D_2R_{spi}$ (Fig. 3a, f). Resultantly, Phe189[5.38] and Ser193[5.42] contacted risperidone in $D_2R_{ris}$, though these residues did not contact haloperidol in $D_2R_{hal}$ (Fig. 3f). The shift of TM5 observed in $D_2R_{ris}$ and $D_2R_{hal}$ is likely inhibited in $D_2R_{spi}$ by steric contact between the extracellular end of TM5 and ECL2. Indeed, Phe189[5.38] in $D_2R_{ris}$ and $D_2R_{hal}$ occupied a similar position with Ile184[45.52] in $D_2R_{spi}$ in the ligand-binding pocket (Fig. 3f). The shift can also be affected by the I122A[3.40] mutation introduced to stabilize the receptor in $D_2R_{ris}$ and $D_2R_{hal}$[22,23]. In $D_2R_{ris}$ and $D_2R_{hal}$, the Cβ atom of Ala122[3.40] was in contact (3.8 Å distance) with the carbonyl oxygen atom of Ser197[5.46] (Supplementary Fig. 7a); when the side chain of Ala122[3.40] was replaced by isoleucine using Coot[45], the resulting side chain formed steric contacts (less than 3.0 Å distance) with the surrounding residues, including Ser197[5.46] and Pro201[5.50] on TM5, and with risperidone or haloperidol in any of the seven allowed side-chain rotamers for isoleucine (Supplementary Fig. 7b).

Unlike inactive state structures, $D_2R_{bro}$ showed a typical active state conformation in the microswitches and the seven helical bundles (Supplementary Fig. 5). Conformations of ECL2 and the extracellular end of TM5 in $D_2R_{bro}$ were more similar to those of $D_2R_{spi}$ than those of $D_2R_{ris}$ and $D_2R_{hal}$. (Fig. 3b, f). Trp100[23.50] of $D_2R_{bro}$ existed at a similar position with those in $D_2R_{spi}$ and $D_2R_{hal}$ but with a different side-chain rotamer (Fig. 3b). EBP in $D_2R_{spi}$ was not observed in $D_2R_{bro}$ (Fig. 3e).

**Comparison with 5-HT$_{2A}$R$_{ris}$ and 5-HT$_{2C}$R$_{rit}$.** The conformations of the extracellular end of TM5, the conserved Trp[23.50] on ECL1, the C-terminal segment of ECL2, and the disulfide bridge between ECL2 and TM3 in 5-HT$_{2A}$R$_{ris}$ and 5-HT$_{2C}$R$_{rit}$ were similar to those of $D_2R_{spi}$ (Fig. 4a, b and Supplementary Fig. 6). On the C-terminal segment of ECL2, the Leu228[45.51] and Leu229[45.52] residues of 5-HT$_{2A}$R contacted risperidone and zotepine, respectively[26], and the Val208[45.52] residue of 5-HT$_{2C}$R interacted with ritanserin[27]. The high conservation of the residues in the ligand-binding pocket (Supplementary Table 4) and the structural similarity of 5-HT$_{2A}$R$_{ris}$, 5-HT$_{2C}$R$_{rit}$, and $D_2R_{spi}$ explain why antipsychotics often bind to $D_2R$, 5-HT$_{2A}$R, and 5-HT$_{2C}$R with high affinity.

Phe110[3.28] and Trp90[2.60] forming EBP in $D_2R_{spi}$ corresponded with Trp151[3.28] and Val130[2.60] in 5-HT$_{2A}$R and Trp130[3.28] and Leu109[2.60] in 5-HT$_{2C}$R, respectively. In $D_2R$, W90L[2.60] reduced the antagonist activity more than F110W[3.28] and similarly with W90L[2.60]/F110W[3.28] (Supplementary Table 3). Spiperone shows a high affinity to $D_2R$ and 5-HT$_{2A}$R but a low affinity to 5-HT$_{2C}$R[25]. A potential explanation of this difference in affinity is that while Trp151[3.28] in 5-HT$_{2A}$R$_{ris}$ can be flipped to form the EBP without any steric hindrance, Trp130[3.28] in 5-HT$_{2C}$R$_{rit}$ is challenging to flip because of the steric contact with Leu109[2.60]

(Fig. 4c, d). Thus, the EBP of $D_2R_{spi}$ and the putative EBP of 5-HT$_{2A}$R$_{ris}$ may contribute to spiperone's higher selectivity for these receptors than for 5-HT$_{2C}$R. The binding mode of spiperone in the EBP may be useful for designing selective $D_2R$ and 5-HT$_{2A}$R antipsychotics.

A unique side-extended cavity was previously observed in the structure of 5-HT$_{2A}$R between TM4 and TM5 that was suggested to contribute to the binding site of 5-HT$_{2A}$R-selective drugs[26]. $D_2R_{spi}$ did not possess the side-extended cavity between TM4 and TM5 (Fig. 4e).

**Comparison with D$_3$R$_{eti}$ and D$_4$R$_{nem}$.** The conformation of $D_2R_{spi}$ was similar to that of $D_3R_{eti}$ and $D_4R_{nem}$, except for the extracellular half of TM6 (Fig. 5a, b). The conformation of the C-terminal segment of ECL2 in $D_2R_{spi}$ was almost identical to that of $D_3R_{eti}$ and $D_4R_{nem}$ (Fig. 5a, b). On ECL2, Ile183[45.52] in $D_3R_{eti}$ and Leu187[45.52] in $D_4R_{nem}$ contacted eticlopride[20] and nemonapride[21], respectively, similar to Ile184[45.52] in $D_2R_{spi}$, which contacts spiperone. A previous study showed that eticlopride and nemonapride bind just above the bottom hydrophobic cleft in $D_3R_{eti}$ and $D_4R_{nem}$[22] (Supplementary Fig. 8a), but this is different from the binding of spiperone in $D_2R_{spi}$, risperidone in $D_2R_{ris}$, and haloperidol in $D_2R_{hal}$. To interact with these benzamide antipsychotics, the extracellular half of TM6 exhibited a greater tilt toward TM3 in $D_3R_{eti}$ and $D_4R_{nem}$ than in $D_2R_{spi}$, $D_2R_{ris}$, and $D_2R_{hal}$ (Fig. 5a, b and Supplementary Fig. 8b). Thus, His349[6.55] in $D_3R_{eti}$ and His414[6.55] in $D_4R_{nem}$ interacted with eticlopride and nemonapride, respectively, whereas no contact was made between spiperone and His393[6.55] in $D_2R_{spi}$ (Supplementary Fig. 8b). Due to the large tilt of TM6, the distances between Cys[3.36] and Phe[6.52] in $D_3R_{eti}$ and $D_4R_{nem}$ were approximately 1.5 and 2.0 Å closer, respectively, than in $D_2R_{spi}$ (Fig. 5a, b and Supplementary Fig. 8b). Together, the side-chain flip of Cys118[3.36] and the tilt of TM6 resulted in the closure of the bottom hydrophobic cleft in $D_3R_{eti}$ and $D_4R_{nem}$ (Fig. 5a, b and Supplementary Fig. 8a, b). Conversely, the insertion of spiperone's fluorophenyl ring between Cys[3.36] and Phe[6.52] created the bottom hydrophobic cleft and inhibited the large tilt of TM6 in $D_2R_{spi}$. $D_2$-class receptors show high affinities for eticlopride, nemonapride, spiperone, and risperidone[25], and display conserved residues in the ligand-binding pocket. Therefore, it is likely that these receptors show a similar conformation with $D_3R_{eti}$ and $D_4R_{nem}$ when they bind to benzamide antipsychotics and with $D_2R_{spi}$, $D_2R_{ris}$, and $D_2R_{hal}$ when they bind to butyrophenone or pyridopyrimidine antipsychotics (Supplementary Fig. 8c).

Spiperone and nemonapride show high affinities for $D_2$-class receptors[25]. In $D_3R$, the putative EBP was closed by the side chain of Phe106[3.28] (Fig. 5c). If the Phe106[3.28] is flipped, $D_3R$ can form an EBP similar to that of $D_2R_{spi}$ with the conserved residues around this region. The EBP in $D_2R_{spi}$ was observed at a position similar to that of the EBP in $D_4R_{nem}$ that binds the phenyl ring of nemonapride[21] (Fig. 5d), although the contact residues were not conserved between $D_2R$ and $D_4R$ (Supplementary Table 4).

## Discussion

$D_2R_{spi}$ was observed to differ substantially from the other inactive state structures of $D_2R$, $D_2R_{ris}$, and $D_2R_{hal}$, especially in ECL2, forming the entrance part of the ligand-binding pocket. This suggests that ECL2 is highly dynamic in the inactive state of $D_2R$. The residues on ECL2 in $D_2R$ have been mapped by the substituted-cysteine accessibility method[46]. In that study, the sulfhydryl groups of I183C[45.51] and I184C[45.52] reacted with charged sulfhydryl-specific reagents, indicating that these residues are water-accessible. The binding of N-methylspiperone to I183C[45.51] and I184C[45.52] was inhibited by the reaction with

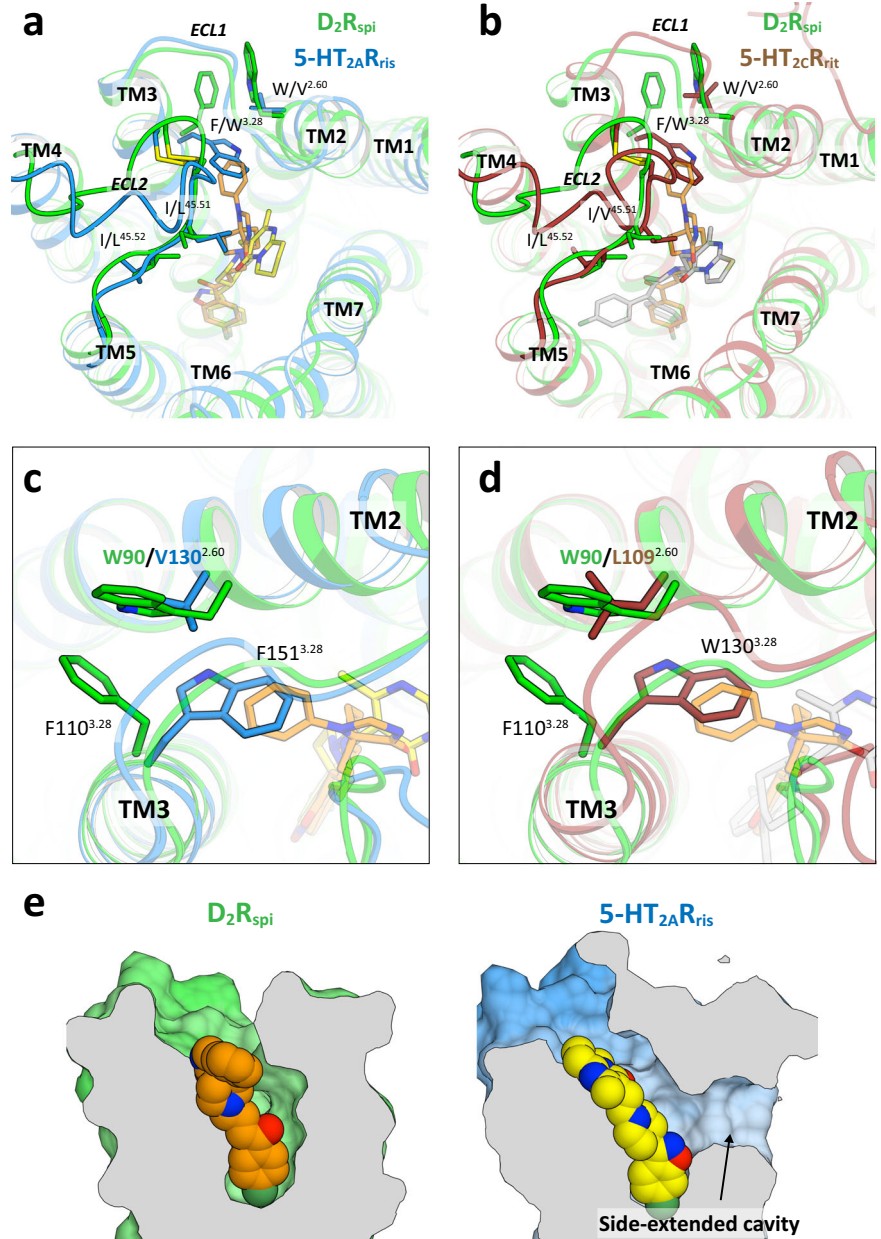

**Fig. 4 Comparison of $D_2R_{spi}$ and 5-HT$_2$ receptors.** Extracellular view of the superpositions of $D_2R_{spi}$ and either 5-HT$_{2A}R_{ris}$ (**a**) or 5-HT$_{2C}R_{rit}$ (**b**). **c** The EBP of $D_2R_{spi}$ and the corresponding part of 5-HT$_{2A}R_{ris}$. **d** The EBP of $D_2R_{spi}$ and the corresponding part of 5-HT$_{2C}R_{rit}$. **e** Vertical cross sections of $D_2R_{spi}$ and 5-HT$_{2A}R_{ris}$. In this figure, $D_2R_{spi}$ (green), 5-HT$_{2A}R_{ris}$ (blue), 5-HT$_{2C}R_{rit}$ (brown), spiperone (orange), risperidone (yellow), and ritanserin (gray) are shown.

sulfhydryl-specific reagents, indicating that Ile183[45.51] and Ile184[45.52] were directed toward the ligand-binding pocket. Ile183[45.51] is less likely to contact the ligand because N-methylspiperone binding to I183C[45.51] was inhibited only by a bulkier sulfhydryl-specific reagent, MTSET. I184C[45.52] reduced the affinities for nemonapride and N-methylspiperone, suggesting that Ile184[45.52] contacts nemonapride and N-methylspiperone. These results are consistent with the conformation of ECL2 observed for $D_2R_{spi}$ but not with that of $D_2R_{ris}$ or $D_2R_{hal}$.

The difference of the ECL2 conformation can be caused by the bound ligand, although $D_2R_{ris}$ and $D_2R_{hal}$ show very similar ECL2 conformation. Indeed, ECL2 in 5-HT$_{2A}$R moves slightly to bind a different ligand[26]. An MD simulation study also suggested these dynamics of the ECL2 of $D_2R$[47] and reported that the helical conformation of ECL2 observed in $D_2R_{ris}$ tended to unwind toward an extended conformation, similar to that of $D_3R_{eti}$,

regardless of the bound ligand, including spiperone, risperidone, or eticlopride[47]. The unwinding involves a drastic rearrangement of the side chain of Ile183[45.51], dissociating from a hydrophobic pocket. The study also suggested that the ECL2 conformation observed in $D_2R_{ris}$ represents a higher energy state than the extended conformation. Considering the structural similarity between $D_2R_{spi}$ and $D_3R_{eti}$, the conformation of ECL2 in $D_2R_{spi}$ likely corresponds to a lower energy state conformation.

Currently, there is a need for novel, safer antipsychotics that bind selectively to 5-HT$_{2A}$R and $D_2R$. In this study, we revealed that the ligand-binding pocket of $D_2R$ forms more than two different conformations in the inactive state. Moreover, we showed that the EBP in $D_2R_{spi}$ and the putative EBP in 5-HT$_{2A}R_{ris}$ could be used as the binding site for selective atypical antipsychotics. $D_2R_{hal}$ was used for the structure-based discovery of selective ligand[23]. The use of multiple different conformations

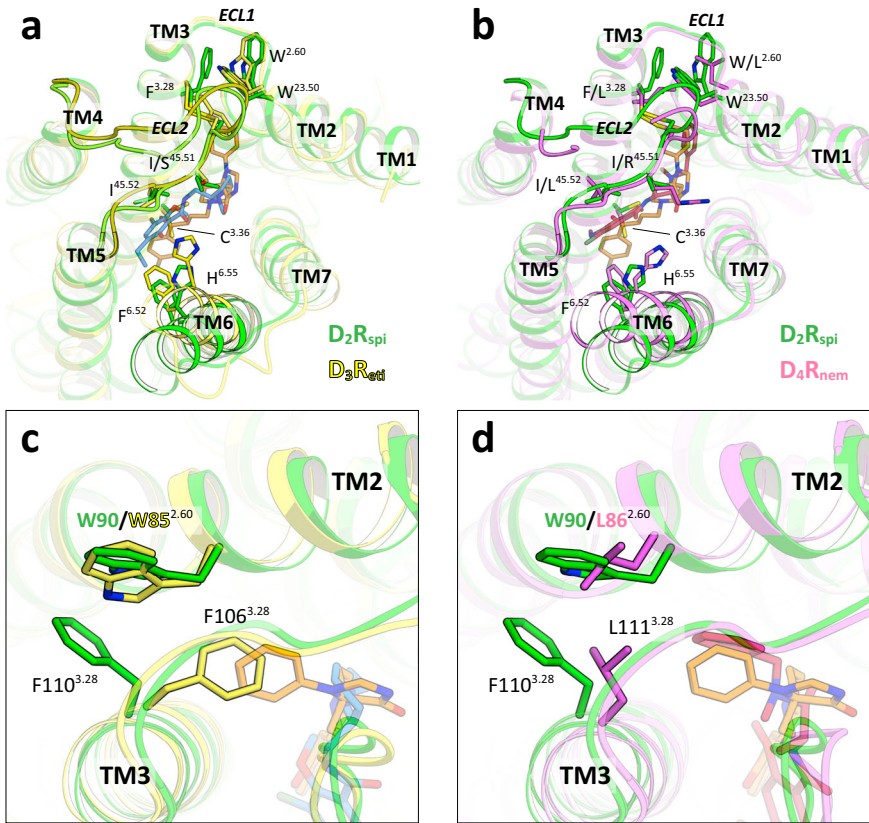

**Fig. 5 Comparison of $D_2R_{spi}$, $D_3R_{eti}$ and $D_4R_{nem}$.** Extracellular view of the superpositions of $D_2R_{spi}$ and either $D_3R_{eti}$ (**a**) or $D_4R_{nem}$ (**b**). **c** The EBP of $D_2R_{spi}$ and the corresponding part of $D_3R_{eti}$. **d** Superposition of the EBP of $D_2R_{spi}$ and $D_4R_{nem}$. In this figure, $D_2R_{spi}$ (green), $D_3R_{eti}$ (yellow), $D_4R_{nem}$ (pink), spiperone (orange), eticlopride (blue), and nemonapride (red) are shown.

in the structure-based design instead of a single conformation clearly increased the possibility of finding high-affinity compounds. Together with $D_2R_{ris}$, $D_2R_{hal}$, and $5-HT_{2A}R_{ris}$, the structure of $D_2R_{spi}$ can be utilized for a rational, structure-based design of new antipsychotics with low side effects.

## Methods

**Protein engineering for structure determination.** The coding sequence of human $D_2R$ (UniProt ID P14416) was synthesized by TAKARA Bio. $D_2R$ was stabilized by removing the N-terminal 34 residues, by introducing two mutations ($S121K^{3.39}$ and $L123W^{3.41}$)[30,48] and by replacing ICL3 ($Lys221^{5.70}$ to $Leu363^{6.25}$) with the loop-modified cytochrome $b_{562}IIG$ mutant ($D_2R$-mbIIG $S121K^{3.39}$/$L123W^{3.41}$)[29]. mbIIG contains four mutations (M7W, R98I, H102I, and R106G), and residues 41–65 have been replaced with the Gly-Ser-Gly-Ser-Gly linker to increase thermostability and reduce conformational variation. All the constructs were prepared by high-throughput fluorescent-based screening in *Saccharomyces cerevisiae*. A 30 ng of SmaI-linearized plasmid pDDGFP2 and 3 µl of the PCR reaction mixtures were co-transformed into *S. cerevisiae* strain FGY217[49]. Transformants harboring the plasmid encoding the receptor were selected on an agar plate without uracil [0.192% (w/v) yeast synthetic dropout media without uracil (Sigma), 0.67% (w/v) yeast nitrogen base without amino acids (BD), 2% (w/v) agar and 2% (w/v) glucose]. The *S. cerevisiae* transformant was cultured in 5 ml of a medium [0.192% (w/v) yeast synthetic dropout media without uracil, 0.671% (w/v) yeast nitrogen base without amino acids and 2% (w/v) glucose] at 30 °C for 24 h. The generated plasmid encoding the receptor was isolated from *S. cerevisiae* with the Miniprep Kit (Qiagen) by disrupting cells with 0.5 mm glass beads[50]. The construct was subcloned into the pFastBac1 vector (Invitrogen), with a C-terminus tobacco etch virus (TEV) protease cleavage site, green fluorescent protein (GFP), and an octa-histidine tag.

**Protein expression and purification.** The stabilized $D_2R$ was expressed in Sf9 cells using a Bac-to-Bac baculovirus expression system (Invitrogen). The Sf9 cells were infected at $1.5 \times 10^6$ cells/ml, at a multiplicity of infection (MOI) of 0.05, and were harvested 84 h later. Cell pellets were resuspended with hypotonic buffer (10 mM HEPES, pH 7.5, 20 mM KCl, and 10 mM $MgCl_2$) and were repeatedly washed and centrifuged in high osmotic buffer (10 mM HEPES, pH 7.5, 1 M NaCl, 20 mM KCl,

and 10 mM $MgCl_2$) containing EDTA-free complete protease inhibitor cocktail (Roche) to purify the cell membranes. The purified membranes were solubilized for 2 h at 4 °C in solubilization buffer (50 mM HEPES, pH 7.5, 500 mM NaCl, 1% (w/v) n-dodecyl-ß-D-maltopyranoside (DDM, Anatrace), 0.2% (w/v) cholesteryl hemisuccinate (CHS, Sigma-Aldrich), and 20% (v/v) glycerol) supplemented with 2 mg/ml iodoacetamide (Wako Pure Chemical Industries, Ltd), 200 µM spiperone (Sigma-Aldrich), and the protease inhibitor cocktail. Insoluble materials were removed by centrifugation, and the supernatants were incubated with TALON metal affinity resin (Clontech) for 10 h at 4 °C. The resin was washed with 10 column volumes (CV) of wash buffer I (50 mM HEPES, pH 7.5, 500 mM NaCl, 10% (v/v) glycerol, 0.05% (w/v) DDM, 0.01% (w/v) CHS, 20 mM imidazole, 10 mM $MgCl_2$, 8 mM ATP, and 100 µM ligand) and 10 CV of wash buffer II (50 mM HEPES, pH 7.5, 500 mM NaCl, 10% (v/v) glycerol, 0.05% (w/v) DDM, 0.01% (w/v) CHS, 20 mM imidazole, and 100 µM ligand). The protein was eluted in 4 CV of elution buffer (50 mM HEPES, pH 7.5, 500 mM NaCl, 10% (v/v) glycerol, 0.05% (w/v) DDM, 0.01% (w/v) CHS, 200 mM imidazole, and 100 µM ligand) and concentrated to 2.5 ml with a 100-kDa molecular weight cutoff Amicon Ultra-15 concentrator (Millipore). The imidazole was removed using a PD-10 column (GE Healthcare). The desalted protein was loaded onto Ni-NTA Superflow resin (Qiagen) and incubated for 10 h. The resin was washed with 10 CV of Ni wash buffer (50 mM HEPES, pH 7.5, 500 mM NaCl, 10% (v/v) glycerol, 0.03% (w/v) DDM, 0.006% (w/v) CHS, 20 mM imidazole, and 100 µM ligand) and eluted with 3 CV of Ni elution buffer (50 mM HEPES, pH 7.5, 500 mM NaCl, 10% (v/v) glycerol, 0.03% (w/v) DDM, 0.006% (w/v) CHS, 400 mM imidazole, and 100 µM ligand). The imidazole was removed using a PD-10 column, and the sample was then incubated with His-tagged TEV protease (expressed and purified in-house) for 10 h. TEV protease, cleaved His-tagged GFP and uncleaved protein were removed by passing the suspension through Ni Sepharose High Performance resin (GE Healthcare).

**Antibody generation.** All the animal experiments conformed to the guidelines of the Guide for the Care and Use of Laboratory Animals of Japan and were approved by the Kyoto University Animal Experimentation Committee (approval no. Med-kyo16043). As the antigen, we used a stabilized $D_2R$ ($D_2R$-mbIIG $S121K^{3.39}$/$L123W^{3.41}$). Purified antigen was reconstituted into liposomes containing chicken egg yolk phosphatidylcholine (Avanti) and monophosphoryl lipid A (Sigma-Aldrich). MRL/lpr mice were immunized three times at two-week intervals with 0.1 mg of the proteoliposome $D_2R$ antigen. Single cells were harvested from mice spleens and were fused with NS-1 myeloma cells. To select antibodies that

recognized the 3D structure of human $D_2R$, we performed a multi-step screening method, using $D_2R$-i3d, which lacks residues of the N-terminal and ICL3, at each step, which included liposome-ELISA, denatured ELISA, and fluorescence size-exclusion chromatography. The collected clones were evaluated using a Biacore T100 protein interaction analysis system (GE Healthcare) and were subsequently isolated by limiting dilution to establish monoclonal hybridoma cell lines. The resulting immunoglobulin-G (IgG3089) was purified with HiTrap Protein G HP (GE Healthcare) followed by proteolytic cleavage with papain (Nacalai Tesque). The Fab fragment (Fab3089) was then purified by size-exclusion chromatography (Superdex 200 10/300 GL, GE Healthcare) and affinity chromatography with a Protein A Sepharose 4 Fast-Flow column (GE Healthcare). The sequence of Fab3089 was determined via standard 5′-RACE using total RNA isolated from hybridoma cells.

**Crystallization**. The $D_2R$–Fab3089 complex was prepared by mixing the purified $D_2R$-mbIIG S121K[3.39]/L123W[3.41] and Fab3089 at a molar ratio of 1:1.2 for 1 h on ice. The mixture was injected onto a Superdex 200 10/300 GL column (GE Healthcare), and the fractions containing the complex were concentrated to approximately 30 mg/ml with a 50-kDa molecular weight cutoff Amicon Ultra-15 concentrator (Millipore). The $D_2R$–Fab3089 in complex with spiperone was reconstituted in LCP by mixing approximately 30 mg/ml protein solution with monoolein and 10% w/w cholesterol at a volume ratio of 2:3 (protein:lipid) using two 100-μl Hamilton syringes and a syringe coupler. One syringe and a coupler were then removed, and a cleaning wire was inserted into the protein-laden LCP in the other syringe[51,52]. Approximately 10 μl of the protein-laden LCP was extruded from the syringe with the wire and was soaked in a 0.6-ml tube filled with pre-cipitant solution (0.1 M Tris-HCl, pH 8.0, 0.1 M $CH_3COOLi$, 28–32% PEG400, 5% dimethyl sulfoxide, 0.01 M ATP, and 1 mM spiperone) and incubated at 20 °C. Microcrystals appeared after 2 days, growing to a maximum size of $20 \times 2 \times 2$ μm$^3$ within a week.

**Data collection using an X-ray free-electron laser**. The data were collected at beamline BL3 of the SPring-8 Angstrom Compact Free-Electron Laser (SACLA)[53] (Hyogo, Japan) by the serial femtosecond crystallography technique using $1.5 \times 1.5$ μm$^2$ microbeams focused by Kirkpatrick–Baez mirrors[54] with a short-working-distance octal multiport CCD detector with eight sensor modules[55]. The data were collected at 7 keV with a pulse duration of approximately 10 fs and a repetition rate of 30 Hz. To inject the microcrystals, LCP was loaded into a sample cartridge through a needle connected to a syringe.

After centrifugation at 2000g for 10 s, the cartridge was mounted in a high-viscosity micro-extrusion injector with a nozzle diameter of 100 μm[51,52,56]. The injector was set in a chamber filled with helium gas in the Diverse Application Platform for Hard X-ray Diffraction in SACLA (DAPHNIS) set-up[57] and was maintained at a constant 20 °C. A total volume of 60 μl of LCP was injected at a flow rate of 420 nl/min. Data collection was guided by a real-time data processing pipeline[58] based on Cheetah[59]. Data processing and indexing were performed using CrystFEL 0.8.0[60,61] and XGANDALF (https://onlinelibrary.wiley.com/iucr/doi/10.1107/S2053273319010593), respectively. The total number of collected, hit, and indexed images were 351,326, 11,373, and 9,464, respectively. The dataset was merged by process_hkl in the CrystFEL suite, without scaling. A per-image resolution cutoff was applied by using the –push-res = 1.2 option to account for variations in the crystal quality.

**Structure determination and refinement**. The structure was determined by molecular replacement with Phaser[62] software using the structures of the trans-membrane region of $D_3R$ (PDB ID: 3PBL), cytochrome b$_{562}$RIL (PDB ID: 1M6T), and the Fab fragment (PDB ID: 1NGZ) as the search models. Refinements were performed using phenix.refine[63] in reciprocal space against experimental structure factors, followed by manual examination and rebuilding of the refined coordinates in Coot[45]. Fourteen TLS groups were used that were chosen by the phenix.find_tls_groups tool[63]. Spiperone was modeled using $2F_o$-$F_c$ map, $F_o$-$F_c$ map, and polder map[64] (Supplementary Fig. 4d). Statistics for the data collection and refinement are shown in Table 1. The Ramachandran statistics analyzed using MolProbity[65] were as follows: 98.0% in the favored region, 2.0% allowed, with no outliers. Figures were prepared using Cuemol (http://www.cuemol.org/) and PyMOL (https://www.pymol.org/).

**Radioligand-binding assay**. The mutants were prepared using the primers listed in Supplementary Table 5. The radioligand-binding assay was performed using HEK293 cell or Sf9 cell membranes that expressed the receptor. The wild-type or a mutant $D_2R$ was transfected with a pCAGGS plasmid into HEK293 cells using a FuGENE HD transfection reagent (10 μg of plasmid, 50 μl of FuGENE HD solution per 10-cm culture dish). The membranes were prepared as described in the "Protein expression and purification" section. The protein concentration of the membrane was determined by the bicinchoninic acid (BCA) method (Thermo Fisher Scientific) with bovine serum albumin as a standard. The membranes were stored at –80 °C until use. All the experiments were performed in triplicate (with independent expressions) in a total volume of 200 μl. The membranes were dispersed with binding assay buffer (50 mM Tris-HCl, pH 7.5, and 150 mM NaCl). Then, 0.5–5 μg of the membranes were incubated for 2 h at room temperature with

[3H]-spiperone (Perkin Elmer) at concentrations of 0.31–20 nM or [3H]-raclopride (Perkin Elmer) at concentrations of 0.20–100 nM. Unifilter-96 GF/B filter plates (Perkin Elmer) were pre-soaked in 0.3% polyethyleneimine (PEI, Nacalai Tesque) for 30 min to reduce non-specific binding. Non-specific binding was determined in the presence of 100 μM spiperone (Sigma) or raclopride (Tocris). For the competition-binding assay, 2 μg of membranes were incubated for 2 h at room temperature with 30 nM of [3H]-raclopride and unlabeled eticlopride at concentrations ranging from 0.01 nM to 1.0 μM. Samples were harvested with Unifilter-96 GF/B filter plates and the unbound ligand was washed three times with distilled water using a FilterMate Cell Harvester system (Perkin Elmer). After adding 20 μl of MicroScint-20 (PerkinElmer), the bound [3H]-spiperone or [3H]-raclopride was quantified with a MicroBeta2 scintillation counter (PerkinElmer). The data were analyzed by nonlinear curve fitting using GraphPad Prism 5 software. To determine Kd value of the receptors for spiperone, we used the equation accounting for ligand depletion. Binding data are reported as the mean ± SEM.

**TGFα shedding assay**. The antagonist activity of spiperone for the mutant $D_2Rs$ was determined by the TGFα shedding assays[32]. Briefly, a pCAGGS plasmid encoding the human wild-type or a mutant $D_2R$ (full-length, untagged), together with pCAGGS plasmids that encoded the chimeric $Gα_{q/i3}$ subunit and alkaline phosphatase-tagged TGFα (AP-TGFα; human codon optimized), were transfected into HEK293A cells that were negative for mycoplasma contamination (MycoAlert Mycoplasma detection kit, Lonza) by using a PEI transfection reagent (PEI MAX MW 40,000; Polysciences). The chimeric $Gα_{q/i3}$ subunit comprises the $Gα_q$ back-bone and the $Gα_{i3}$-derived 6-amino acid C-terminus, and it couples with $G_i$-coupled $D_2R$, but induces a $G_q$-dependent TGFα shedding response[32]. For each 10-cm culture dish, we used 1 μg of $D_2R$ plasmid, 0.5 μg of $Gα_{q/i3}$ plasmid, 2.5 μg of AP-TGFα plasmid and 25 μl of 1 mg/ml PEI solution. After culturing for one day at 37 °C in a 5% $CO_2$ incubator, the transfected cells were harvested by trypsinization, washed once with Hank's balanced salt solution (HBSS) containing 5 mM HEPES (pH 7.4), and resuspended in 30 ml of the HBSS-containing HEPES. The cell suspension was seeded in a 96-well culture plate ("cell plate") at a volume of 80 μl per well and incubated for 30 min in the $CO_2$ incubator. To determine the antagonist activity of spiperone or risperidone, cells were pretreated with 3.2-fold-titrated concentration of the antagonists (final concentrations of 32 pM–1 μM for spiperone or 320 pM–10 μM for risperidone; 10 μl per well) for 5 min and stimulated with dopamine (final concentration of 1 μM; 10 μL per well). To determine the agonist activity of dopamine, vehicle (10 μL per well) were predispensed before cell seeding and 3.2-fold-titrated concentration of dopamine (final concentrations of 1 nM–32 μM; 10 μl per well) was added to the cells. For all the experiments, the compounds were diluted in 0.01% bovine serum albumin- and HBSS-containing HEPES. After incubation with dopamine for 1 h, the cell plate was spun at $190 \times g$ for 2 min and conditioned medium (CM; 80 μl per well) was transferred to an empty 96-well plate ("CM plate"). Alkaline phosphatase reaction solution (10 mM p-nitrophenylphosphate, 120 mM Tris-HCl, pH 9.5, 40 mM NaCl, and 10 mM $MgCl_2$) was dispensed into the cell plates and CM plates (80 μl). The absorbance of the plates at 405 nm was measured using a microplate reader (SpectraMax 340 PC384, Molecular Devices) before and after incubation for 1 or 2 h at room temperature. Ligand-induced AP-TGFα release was obtained by calculating AP activity in conditioned media and subtracting a vehicle-treated spontaneous AP-TGFα signal[32]. Using Prism 8 software (GraphPad Prism), the AP-TGFα release signals were fitted with a four-parameter sigmoidal concentration–response curve, from which $EC_{50}$ or $IC_{50}$ and $E_{max}$ values were obtained. Negative logarithmic values of $EC_{50}$ ($pEC_{50}$) were used to calculate the mean and SEM.

The equilibrium dissociation constant ($K_B$) was calculated for each experiment performed in parallel from the $IC_{50}$ values (for spiperone and risperidone), an $EC_{50}$ value (for dopamine), a Hill slope ($K$, for dopamine), and the tested concentration of dopamine ($A$; 1 μM), as follows[66]:

$$K_B = \frac{IC_{50}}{1 + \left(\frac{A}{EC_{50}}\right)^K} \tag{1}$$

The resulting $K_B$ values were logarithmically transformed and their negative values ($pK_B$) were used to calculate the difference between the $pK_B$ values ($\Delta pK_B$) for a mutant (MT) and the wild-type (WT) receptor, derived from parallelly conducted experiments, as follows:

$$\Delta pK_B = pK_B(MT) - pK_B(WT) \tag{2}$$

Mean and SEM values of the $pK_B$ and the $\Delta pK_B$ values were calculated.

**Reporting summary**. Further information on research design is available in the Nature Research Reporting Summary linked to this article.

## Data availability

The coding sequence of human $D_2R$ is available in UniProt with accession code P14416. The protein coordinate and atomic structure factor have been deposited in the Protein Data Bank (PDB) with accession code 7DFP. The raw diffraction images have been deposited to CXIDB (https://cxidb.org/) with accession code 110. Other data are available from the corresponding authors upon reasonable request. Source data are provided with this paper.

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

## Acknowledgements

This work was supported by grants from the Research Acceleration Program of the JST (S.I.), the X-ray Free-Electron Laser Priority Strategy Program from MEXT (T.S., S.I.), JSPS KAKENHI (Grant Nos. 24370044, 24121715, 26102725, 15H04338, 17K19349, 18H02388, and 20K21392 to T.S.; Grant No. 15K18376 to D.I.; 17K08264 to A.I.; JP19H05777 to S.I.) and the Mitsubishi Foundation (T.S.). This research was also supported by the Platform Project for Supporting Drug Discovery and Life Science Research (Platform for Drug Discovery, Informatics, and Structural Life Science) and the Basis for Supporting Innovative Drug Discovery and Life Science Research (BINDS) from AMED under Grant Numbers JP17am0101070 and JP20am0101079; the PRIME JP17gm5910013 (A.I.) and the LEAP JP17gm0010004 (A.I. and J.A.) from AMED. We thank Kayo Sato, Yuko Sugamura and Ayumi Inoue (Tohoku University) for optimization and technical assistance of the TGFα shedding assay. We thank the Radioisotope Research Center and Medical Research Support Center at the Kyoto University for instrumental support. X-ray crystallographic data were collected at SACLA (Proposal No. 2016B8060, 2017A8019, and 2017B8022). We acknowledge computational support from the SACLA HPC system and the Mini-K supercomputer system.

## Author contributions

D.I. designed constructs, expressed, purified, and crystallized the receptor. D.I. and T.U. generated the antibody. D.I., T.F., and C.M. prepared microcrystals. D.I., T.F., Y.Y., T.T., A.Y., and E.N. collected data at SACLA. T.N. processed the data. K.T. contributed to the beamline operation at SACLA. Y.S. and N.N. prepared mutants. A.I and F.M.N.K. designed, performed, and analyzed the functional assay under J.A.'s supervision. H.A. and K.T.K. performed the binding assay. T.S. designed the constructs and solved and refined the structure as well as supervised the project. D.I, S.I., and T.S. wrote the manuscript. S.I. advised T.S. All authors discussed the results and commented on the manuscript.

## Competing interests

The authors declare no competing interests.
