## [Peer Review File · Nature Communications]

Reviewers' comments:

Reviewer #1 (Remarks to the Author):

The authors present the structure of dopamine D2 receptor bound to the drug spiperone (D2-spi) and a stabilizing Fab fragment, Fab3089. This is the second structure determined for D2 receptor and the fourth for a dopamine receptor, after D2-risperidone (D2-ris) from the Roth lab (2018), and dopamine receptor sub-types D3 and D4 bound to different ligands (2010, 2017). D2 is an important therapeutic target in context of unwanted side effects of antipsychotics. While typically general interest in subsequent structures is diminished, in this present case, in principle it could be warranted given the authors outline perceived shortcomings of the original D2-risperidone structure in great detail and call into question its usefulness for structure-based drug design. The most important differences they describe are an alternative conformation of extracellular loop 2 (ECL2) which forms contacts with ligand in their structure, but not that from Roth lab, and binding pocket differences around the so-called PIF motif that has been mutated in the structure from Roth lab.

It is vital, though, that such a comparison to an earlier structure is carried out as carefully as possible, and that the later study indeed corrects crucial mistakes of the earlier study. In my opinion, the authors fall short of this important goal in several ways, and their study, in the current form, has severe technical and conceptual flaws which will be outlined below.

Minor concerns, requests and comments.

The authors mutate S121K, among other modifications, to stabilize receptor for crystallographic studies. While the origin of this mutation is attributed, it should be explicitly mentioned in the main text that this is a mutation of the allosteric sodium site of GPCRs that mimicks the presence of sodium and therefore stabilizes the inactive / antagonist-bound receptor state, as described in the EP4 structure of Toyoda et al (2018) where a corresponding mutation was used, and also in the M2 structure of Suno et al (2018).

The authors use modified, thermostabilized BRIL as fusion partner. Is its use for GPCR crystallization the first reported case, or has it been used in the past? Was standard BRIL unsuccessful?

It should be explicitly stated in the main text that Fab3089 is a novel Fab developed for this structural study.

The authors should state the overall RMSD between their structure and the structure from Roth lab. They should also compare helical bundle packing with other active and inactive structures and state whether based on helical packing, and also the conformation of microswitches, their structure has been crystallized in active or inactive state.

The authors state (line 145) that "interactions with Trp386 and F390 are essential for binding", although they only present functional data to back up their claim. It would be advisable to not conflate ligand binding with ligand induced activation throughout the manuscript, especially given that residue Trp386 is a known microswitch involved in dynamic response to activation.

The authors further point out a perceived artefact in the earlier study from Roth lab where residue 122, were it the natural isoleucine, would clash with surrounding residues and ligand (line 205).

Their own structure has the natural isoleucine in this position, and those clashes are prevented by backbone adjustments around this site. To determine this clash they choose a cutoff of 3.0 Å for steric contacts – however, their own structure (according to PDB validation and Molprobit server) contains 4-5 examples of closer than 3.0 Å contacts between non-hydrogen atoms, e.g. a 2.15 Å distance between oxygens of residues 109 and 112. These steric clashes should be corrected.

While the overall structure is extremely well refined judging stereochemical figures of merit in Molprobit, it does feel overrefined/overmodelled in many places. In particular, given the low resolution of the structure, there is no experimental evidence (even at generous 0.7 sigma level) in the electron density for a large number of receptor side chains (Leu40, Leu43, Val55, Arg61, Leu65, Val74, Lys101, Lys121, Lys125, Ser148, Lys149, Arg150, Met155, Leu174, Ile184, Phe189, Ile214, Leu216, Lys369, Lys370, Tyr408, Phe433, Lys435, Lys439, His442), some of them in the ligand binding site. All these side chains should be trimmed, and consistent criteria should be employed for the modeling of Fab and fusion partner.

It is apparent from the coordinate file that TLS refinement was used. This should be explicitly stated in the methods. How were the 11 TLS groups chosen? This seems a high number. Have the TLS contributions been summed back into the overall B-factor?

Why was ATP used as additive during GPCR crystallization? Its relatively high concentration (10 mM) is ten-fold molar excess over spiperone ligand; would the authors expect unspecific binding of ATP to receptor at such high concentrations, or does ATP rather serve another role, e.g. as crystallization additive? Was ATP essential for crystallization, or diffraction quality?

The authors state that "spiperone was modelled based on polder map" (line 446). The maps they present in their manuscript (Suppl Fig 1) looks much nicer than what I see in Coot where there is no continuous ligand density (i.e. no density for carbons 18 and 19). Can the authors please comment on why this rather unusual map was used, and clearly state if the ligand-containing complex was refined in reciprocal space against experimental structure factors, or only in real space against polder map?

The ligand geometry shows a large number of bad or questionably angles – was a correct geometry file used for refinement?

The authors should show graphs for binding isotherms and dose-response curves for the data they present in their supplementary tables.

Major concerns, requests and comments.

Radioligand binding was performed in Sf9 cells. The authors should obtain data points in HEK cells (more relevant for human receptor) for comparison. Importantly, the authors only quote a range of affinities from literature for wild-type D2 (Suppl Table 1) that spans a range of three orders of magnitude. While their measured affinity value for crystallized construct falls within this wide affinity range, they should provide their own wild type affinity measurement carried out under comparable conditions as the mutated constructs to judge the true difference in binding between crystal construct and wild type. The authors should also characterize the individual influence of their two point mutations in wild type background (radioligand binding and functional assays). The authors show more comprehensive functional than binding data; however, they don't show functional data for their stabilizing point mutations. The role of these mutations has to be more carefully characterized.

My main concern with this study is that it compares two systems (D2-spi-Fab vs D2-ris) where no two parts of these systems are the same, and draws major conclusions about the resulting observed differences. While it is possible that their receptor construct is closer to the wild-type than that of Roth lab due to a different set of mutations used (although they show no binding or functional validation for that), also the ligands are different, and the authors additionally use a Fab that interacts with the extracellular part of the receptor where a major conformational difference is observed compared to the previous structure. The difference in ECL conformation can be induced by their ligand, e.g. its additional substituent opening up the extended binding pocket that they describe, or the Fab, or both, and it's unclear if this difference is important for drug design, i.e. makes their structure more useful for drug design than the previous structure. It is clear from the structure that Fab stabilizes the ECL conformation, and in part protrudes into the D2-binding site (e.g. residues Phe54 and Tyr55 of Fab chain C), pushing outward D2-TM5. The authors don't provide functional or binding data in presence of Fab, or indeed any characterization of Fab beyond the structure itself – a comparable study (EP4 structure by Toyoda et al 2018) much more carefully characterized the Fab used there. Without any characterization of Fab (binding and functional) the biological relevance of the changes the authors observe can't be estimated.

Can both D2-spi-Fab and D2-ris structures be correct, and observed differences be inherent to the difference of systems studied? – e.g. Roth lab describe no effect of mutating ECL2 residue 184 on binding/kinetics of several ligands, while the present study finds a large effect on receptor function in presence of their (different) ligand. The authors should test the 184A mutation in their binding assay also, not only functional, and they should test effect of this mutation on the risperidone ligand using their assays to exclude assay artefacts. Perhaps the ECL2 is dynamic and does not interact with risperidone, but it interacts with spiperone – they comment that this is unlikely (line 327) based on comparison to structure of 5-HT2A-ris complex structure, but I don't find such a comparison between two classes of receptors very convincing, even though they share ligands, absent corroborating experimental results.

Reviewer #2 (Remarks to the Author):

Comments for Authors:

The presented manuscript 'Structure of the dopamine D2 receptor in complex with the antipsychotic drug spiperone' by So Iwata, Tatsuro Shimamura and colleagues characterizes the structural basis of the dopamine D2 receptor (D2R) in complex with an antipsychotic drug, the D2R-antagonist spiperone. The authors combine XFEL structure data of the D2R-spiperone complex (D2Rspi) and several interesting substitutions of D2R binding pocket as well as a very comprehensive comparison with the available structures of D2R in complex with risperidone (D2Rris) and other Dopamine and Serotonin receptors D3R, D4R and 5-HT2A(C)R. The D2Rspi structure shows some decisive differences in the ligand-binding pocket compared with D2Rris. They also found some explanations for the high specificity of spiperone to the D2R and 5-HT2AR. The structural data look well refined in relation to the limitation in the resolution of 3.2Å. Also all other included data are technically well done.

The value of new information in this study is given by the new and extensive structural comparisons.

The new structure information of the D2Rspi structure is, however, somewhat limited and for my feeling not substantially more or less artificial than in the already known crystal structure D2Rris (Nature 2018). In addition, many mutants in D2R have already been investigated before. The manuscript would benefit a lot from (a bit) shortening the text (e.g. combining structure comparisons) and making the images clearer, and above all not talking down about the other D2Rris structure in every section.

Nevertheless, I think that this study and the new data can contribute to a better understanding and to new approaches for D2R-ligand development.

Remarks:

- (line 76-77). The authors could highlight/label the PIF motif in figure 2a. (line 154/156) Is there really a direct interaction between Ile122 of PIF and spiperone? What is the exact distance?

- (line 83-84): The sentence is misleading. Spiperone binds to several other receptors (D3R, D4R etc.)

- The authors could improve Figure 1. The blue color in a and c and the depth of field of the receptor image are not so catchy (with a better positioning of the labels).

- (line 77-81): The authors should explain this small paragraph in a better way and at more detail. Maybe a visualization with a new supplemental figure would help. For example, ECL1 has in D2R, D3R and D4R almost the identical conformation. Only Trp100 in D2R looks different and is directed towards the binding pocket. On the other hand, ECL2 is completely different in D2R compare with D3R and D4R. In both structures, D2Rris and D2Rspi the residue Trp100 is important for the antagonist binding. I didn't understand why the D2Rris structure is here more artificial. It is a different antagonist and maybe induce a different binding mode. The argumentation here is too far-reaching and both structures are biased by their crystallization constructs.

- (line 97-98): The authors should determine the affinity (Kd) of the wild-type D2R themselves (Table 1). The affinity of the published (in 1990) wild-type D2R to spiperone is different compared to both crystallization constructs.

- The comparison of D2Rris and D2Rspi shows a clear 2 Å motion of TM6. How exactly is this motion compared with D3Reti and D4Rnem. The outward motion of the TM6 at the binding pocket also seems to be significantly different in D3R and D4R.

- (line 125-): The contact to of Ile183 seems to be not important (SI Table2) which is absolute in line with the D2Rris structure (no contact). Maybe the ECL2 has here a different functional interaction in both cases, D2Rris vs. D2Rspi, (or as an intermediate conformation of ECL2 in D2Rris). For my feeling the variations of ECL2 per se is the interesting point here.
- A very important difference is the different orientation of F110 in D2Rris vs. D2Rspi. In D2Rspi F210 is facing to the solvent, but this is clear due sterically hindrance of the phenyl ring from spiperone. In risperidone this phenyl ring does not exist, so the interaction between the two antagonists a D2R must be different at this point. That is in very interesting feature. What happens if this is mutated to Trp (as in 5-HT2a-R).
- (line 193-194): The sentence is very misleading. Ligand binding is not a static process.
- (line 207-210): The sentence clearly goes too far. As asked above, how is the contact distance in the D2Rspi structure from Ile122 to spiperone? Risperidone is also clearly a different ligand than spiperone. Both bind simply differently in some details (different head group like flourphenyl etc). And as a remark, as the authors have explained very nicely in their own publication (Kimura et al. NSMB 2019) about the 5-HT2AR: '5-HT2ARris, the entrance of the ligand-binding pocket between TM7 and ECL2 is wider by up to 2.2 Å than that in 5-HT2ARzot, which is essential for binding to the tetrahydropyridopyrimidinone ring of risperidone'. The different ligands deform the binding pocket also simply different.
- (line 219-221): The sentence is misleading. The pockets are obviously different in more than one amino acid residue (F/W3.28). This is nicely shown in figure 4e. The authors could name other strong differences such as L2.64, which is really different from the other receptors, 5-HT2ARzot and 5-HT2ARris.
- (line 233-254): This is one potential explanation. The authors should put it that way. I am not so convinced because the receptors are very dynamic. The D2R pairs are F and W and in the 5-HT structures W and L. They also could investigate a double mutation F130W3.28 and W90L2.60 for a comparison with the 5-HT structures. But it is surly one crucial site for ligand selectivity.
- (line 266-268): This is also true for D2Rris (larger TM6 movement).
- (line 333-335) : To my mind this sentence in this form is really unnecessary. Both structures have more or less their justification for a structure-based drug design
- Figures 3b and 4a,b are overloaded and really hard to distinguish. In Figure 4 the same color disturbs for 5-HT2ARris vs D2Rspi.
- The sentence in figure 3 is partially not correct? The text reads as follows '...residues in D2Rspi and the ... in D2Rris are shown in cyan and magenta sticks, respectively. Red arrows indicate the shift of helices in D2Rris with distance relative to D2Rspi.' It should revise to 'Red arrows indicate the shift of helices in D2Rspi with distance relative to D2Rris.'
- The 2fo-fc electron density in supplemental figure 1c is not very nice. I see almost nothing but a blue bolb? The authors should improve this figure significantly.
- A second different view of supplemental figure 1d would be very insightful.

Reviewer #3 (Remarks to the Author):

This manuscript describes a new crystal structure of dopamine D2 receptor (D2R) in complex with the antagonist spiperone (D2Rspi). There is one previous D2R structure in complex with

risperidone (D2Rris). This previous structure required several thermostabilizing mutations, which are not present in the current structure (however, two other mutations were introduced in D2Rspi, see below). The D2Rspi presented in this manuscript is different from the previous D2R structure in several striking ways in or near the ligand binding pocket, in particular in the conformations of ECL1, ECL2 and TM5. The author also compared D2Rspi to other related 5-HT2ARris, 5-HT2CRrit, D3Reti and D4Rnem structures. They identified the structural basis for the high-affinity of spiperone binding for D2R and 5-HT2AR but not for 5-HT2CR, i.e., the residue configurations at positions 3.28 and 2.60. They conclude that the D2R, D3R, and D4R conformations may adapt to different types of antipsychotics. The authors also carried out systematic mutagenesis study to validate the spiperone binding residues identified in the D2Rspi structure.

This new D2Rspi structure together with previous D2Rris, D3Reti and D4Rnem provide important clues for the field to understand the ligand-receptor recognitions for these highly-homologous and therapeutically important receptors. I expect the manuscript to be of high impact because unlike many new structures that reveal only incremental advances, this structure suggests that different antagonists stabilize quite different receptor conformations, which will have major impact not only on virtual screening for new ligands but also on potential insights into biased signaling. While these ideas are only hinted at in the manuscript, the work is well supported by the mutagenesis data, making the results an important contribution to the literature.

I only have a few suggestions,

For the statement in the abstract, "D2Rris exhibits artificial conformation in the ligand-binding pocket owing to a mutation introduced for the stability, limiting efficient development of antipsychotics", the authors need to elaborate why the conformation is artificial. Note that the impact of I122A on binding is relatively small, and the risperidone pose is similar in D2Rris and 5-HT2ARris (see supplementary figure 4), which does not have a mutation at the aligned position (ref 36). I think it remains an open question whether the mutation led to an artificial pose or whether the different drug scaffolds can stabilize different conformations, and this should be discussed. The authors may also want to provide data to support why they believe that D2Rris may limit efficient development of antipsychotics.

A related issue is that the current structure also has thermostabilizing mutations, albeit different. Thus, this structure is subject to the same consideration. Is the structure "artificial" or does it represent a different conformation with a different scaffold that naturally occurs, aided by the mutations. This is likely unknowable but it seems inappropriate to criticize the previous structure without applying the same caveat here, i.e., the impact of the S121K and L123W mutations introduced in the D2Rspi structure. Curiously, the K_d in supplementary table 1 are in nM ranges, while the spiperone affinity has been more often reported in pM range (see literature and compare supplementary tables 1 and 2). If "D2R-bRIL", a NT-truncated and IL3-replaced construct, which is not exactly the background construct of "D2R-mbIIG S121K3.39/L123W3.41" with a different IL3 replacement, has a disrupted spiperone binding already, it is critical to understanding how the mutations impaired binding themselves. Interestingly, an examination of the structure reveals that the sidechain of S121K protrudes in the Na⁺ binding site. As Na⁺ binding may potentially affect the binding of particular ligands as well as the receptor conformation (see Neve et al., Mol Pharm 1991, 2001; Michino et al., Chem Commun 2015), this may be relevant to the arguments in this manuscript. The impact of the S121K and/or L123W need to be tested and compared with wt to supplement the results shown in supplementary table 2. Alternatively, if these two mutants render the receptor functionally inactive, this should be reported and radioligand binding studies should be performed to compare the affinity of spiperone at this mutant and the WT receptor. It would not be unreasonable to also add functional inhibition or binding affinity data on a Na⁺ dependent ligand, such as eticlopride or sulpiride to ascertain the impact of the potential disruption of the Na⁺ site. These data seem important to characterizing the construct used to generate this interesting structure.

On page 18, the first paragraph of discussion, the I183C results from ref 23 seem to be slightly

over-interpreted. I183C was shown to have low accessibility that was only detected by MTSET, so it is less likely to be in direct contact with ligand but is more likely directed toward the ligand-binding pocket. Interestingly, in Supplementary Table 2 of the current manuscript, I183A appears to improve spiperone's affinity by 6 fold. Given that the sidechain of I183 in the crystal structure appears to have been stabilized by a Tyr from Fab3089, the authors need to be more cautious in interpreting the interaction between I183 and spiperone.

Reviewers' comments:

Below we provide our replies to the comments of the reviewers in blue following the reviewers' remarks in black.

Reviewer #1 (Remarks to the Author):

The authors present the structure of dopamine D2 receptor bound to the drug spiperone (D2-spi) and a stabilizing Fab fragment, Fab3089. Theirs is the second structure determined for D2 receptor and the fourth for a dopamine receptor, after D2-risperidone (D2-ris) from the Roth lab (2018), and dopamine receptor sub-types D3 and D4 bound to different ligands (2010, 2017). D2 is an important therapeutic target in context of unwanted side effects of antipsychotics. While typically general interest in subsequent structures is diminished, in this present case, in principle it could be warranted given the authors outline perceived shortcomings of the original D2-risperidone structure in great detail and call into question its usefulness for structure-based drug design. The most important differences they describe are an alternative conformation of extracellular loop 2 (ECL2) which forms contacts with ligand in their structure, but not that from Roth lab, and binding pocket differences around the so-called PIF motif that has been mutated in the structure from Roth lab.

It is vital, though, that such a comparison to an earlier structure is carried out as carefully as possible, and that the later study indeed corrects crucial mistakes of the earlier study. In my opinion, the authors fall short of this important goal in several ways, and their study, in the current form, has severe technical and conceptual flaws which will be outlined below.

We thank you for your careful review of our manuscript. We have answered each of your comments below.

Minor concerns, requests and comments.

The authors mutate S121K, among other modifications, to stabilize receptor for crystallographic studies. While the origin of this mutation is attributed, it should be explicitly mentioned in the main text that this is a mutation of the allosteric sodium site of GPCRs that mimicks the presence of sodium and therefore stabilizes the inactive / antagonist-bound receptor state, as described in the EP4 structure of Toyoda et al (2018) where a corresponding mutation was used, and also in the M2 structure of Suno et al (2018).

We appreciate this suggestion and have added a discussion of this topic in the main text, as described in the EP4 structure paper (page 5, lines 89–91).

The authors use modified, thermostabilized BRIL as fusion partner. Is its use for GPCR crystallization the first reported case, or has it been used in the past? Was standard BRIL unsuccessful?

We designed mBIIG and applied it to D₂R first and then to 5-HT_{2a}R, although the structure paper of 5-HT_{2a}R had already been published. Standard BRIL was unsuccessful (page 5, line 88-89).

It should be explicitly stated in the main text that Fab3089 is a novel Fab developed for this structural study.

Accordingly, we have stated this in the main text (page 6, lines 106–107).

The authors should state the overall RMSD between their structure and the structure from Roth lab.

We thank the reviewer for this advice. Our revised manuscript now states the RMSD values among D₂R structures (Supplementary Table 1). The table clearly shows the difference in ECL2 between D₂R_{spi}, D₂R_{ris}, and D₂R_{hal}.

They should also compare helical bundle packing with other active and inactive structures and state whether based on helical packing, and also the conformation of microswitches, their structure has been crystallized in active or inactive state.

Per this suggestion, we have compared the helical bundle packing and the conformations of microswitches of D₂R_{spi} with those of an inactive conformation (D₂R_{ris}) and an active conformation (D₂R_{bro}) of D₂R. Because D₂R_{ris} contains a mutation in a microswitch (PIF motif) we also compared the conformation of the PIF motif with that of an inactive structure and an active structure of the β₂-adrenergic receptor (Supplementary Fig. 5). These comparisons clearly show that the structure of D₂R_{spi} is in the inactive state (pages 6–7, lines 112–119).

The authors state (line 145) that “interactions with Trp386 and F390 are essential for binding”, although they only present functional data to back up their claim. It would be advisable to not conflate ligand binding with ligand induced activation throughout the manuscript, especially given that residue Trp386 is a known microswitch involved in dynamic response to activation.

We have rewritten the corresponding text not to conflate ligand binding with ligand-induced activation (page 8, lines 160–163).

The authors further point out a perceived artefact in the earlier study from Roth lab where residue 122, were it the natural isoleucine, would clash with surrounding residues and ligand (line 205). Their own structure has the natural isoleucine in this position, and those clashes are prevented by backbone adjustments around this site. To determine this clash they choose a cutoff of 3.0 Å for steric contacts – however, their own structure (according to PDB validation and Molprobity server) contains 4-5 examples of closer than 3.0 Å contacts between non-hydrogen atoms, e.g. a 2.15 Å distance between oxygens of residues 109 and 112. These steric clashes should be corrected.

We have reprocessed the data using the newer programs, thus improving the resolution and the statistics (Table 1). Using these data, we refined the structure and corrected all the steric clashes.

While the overall structure is extremely well refined judging stereochemical figures of merit in Molprobity, it does feel overrefined/overmodelled in many places. In particular, given the low resolution of the structure, there is no experimental evidence (even at generous 0.7 sigma level) in the electron density for a large number of receptor side chains (Leu40, Leu43, Val55, Arg61, Leu65, Val74, Lys101, Lys121, Lys125, Ser148, Lys149, Arg150, Met155, Leu174, Ile184, Phe189, Ile214, Leu216, Lys369, Lys370, Tyr408, Phe433, Lys435, Lys439, His442), some of them in the ligand binding site. All these side chains should be trimmed, and consistent criteria should be employed for the modeling of Fab and fusion partner.

Data reprocessed with newer programs, as above, improved the electron density. The use of a feature-enhanced map also helped to place the side chains. We trimmed the side chains when they showed no electron density, even in the feature-enhanced map. These enhancements have been implemented throughout the revised manuscript.

It is apparent from the coordinate file that TLS refinement was used. This should be explicitly stated in the methods. How were the 11 TLS groups chosen? This seems a high number. Have the TLS contributions been summed back into the overall B-factor?

Indeed, for the refinement of the new structure, we used 14 TLS groups. The TLS groups were chosen by the `phenix.find_tls_groups` tool. We now state these parameters in the Methods section (pages 19–20, lines 418–419).

Why was ATP used as additive during GPCR crystallization? Its relatively high concentration (10 mM) is ten-fold molar excess over spiperone ligand; would the authors expect unspecific binding of ATP to receptor at such high concentrations, or does ATP rather serve another role, e.g. as crystallization additive? Was ATP essential for crystallization, or diffraction quality?

We added ATP as a crystallization additive. The addition of ATP improved the diffraction quality of the crystallization.

The authors state that “spiperone was modelled based on polder map” (line 446). The maps they present in their manuscript (Suppl Fig 1) looks much nicer than what I see in Coot where there is no continuous ligand density (i.e. no density for carbons 18 and 19). Can the authors please comment on why this rather unusual map was used, and clearly state if the ligand-containing complex was refined in reciprocal space against experimental structure factors, or only in real space against polder map?

As shown on the Phenix website, “a polder map is an omit map which excludes the bulk solvent around the omitted region. This way, weak densities, which can be obscured by bulk solvent, may become visible.” Because the $2Fo-Fc$ map and $Fo-Fc$ map were not strong, we also used a polder map to check the conformation of spiperone. We have corrected the sentence as follows: “Spiperone was modeled using $2Fo-Fc$ map, $Fo-Fc$ map, and polder map.” We also have stated in the Methods section that the refinements were performed using `phenix.refine` in reciprocal space against experimental structure factors (page 19, lines 417, page 20, lines 419-420).

The ligand geometry shows a large number of bad or questionably angles – was a correct geometry file used for refinement?

In the revised manuscript, we have corrected the geometry file and used it for refinement.

The authors should show graphs for binding isotherms and dose-response curves for the data they present in their supplementary tables.

Accordingly, we now show graphs of the antagonist assay and the binding assay in Supplementary Figures 2 and 3.

Major concerns, requests and comments.

Radioligand binding was performed in Sf9 cells. The authors should obtain data points in HEK cells (more relevant for human receptor) for comparison. Importantly, the authors only quote a range of affinities from literature for wild-type D2 (Suppl Table 1) that spans a range of three orders of magnitude. While their measured affinity value for crystallized construct falls within this wide affinity range, they should provide their own wild type affinity measurement carried out under comparable conditions as the mutated constructs to judge the true difference in binding between crystal construct and wild type. The authors should also characterize the individual influence of their two point mutations in wild type background (radioligand binding and functional assays). The authors show more comprehensive functional than binding data; however, they don't show functional data for their stabilizing point mutations. The role of these mutations has to be more carefully characterized.

We have successfully established the expression systems in HEK cells for the wild-type, S121K, L123W, and S121K/L123W D₂Rs and determined their binding affinities. As shown in Supplementary Table 2, these mutants possess similar affinities for spiperone to the wild-type receptor. We also performed a shedding assay for these mutants. L123W showed similar antagonist activity to the wild-type. Because S121K stabilizes the inactive state, the antagonist activities of S121K and S121K/L123W could not be determined (Supplementary Table 3). These results suggest that these mutations do not affect spiperone binding.

However, because we were unable to express the crystallized construct in HEK cells, we determined the binding affinity of the crystallized construct using the receptor expressed in Sf9 cells. Additionally, we failed to determine the binding affinity of the wild-type expressed in Sf9 cells due to its low expression. The crystallized construct showed a similar affinity for spiperone to the wild-type, as shown in Supplementary

Table 2. Although the receptors were expressed in non-human cells, it is reasonable to consider that the introduced mutations in the crystallized construct do not affect the spiperone binding.

We noticed that ligand depletion occurred in the ligand-binding assay for spiperone. Therefore, we changed the assay method by increasing the assay volume and using the equation accounting for ligand depletion in GraphPad Prism 5 software. Thus, the K_d value of the crystallized construct in the revised manuscript ($K_d = 0.29$ nM) is different from that in our original manuscript ($K_d = 3.0$ nM).

My main concern with this study is that it compares two systems (D2-spi-Fab vs D2-ris) where no two parts of these systems are the same, and draws major conclusions about the resulting observed differences. While it is possible that their receptor construct is closer to the wild-type than that of Roth lab due to a different set of mutations used (although they show no binding or functional validation for that), also the ligands are different, and the authors additionally use a Fab that interacts with the extracellular part of the receptor where a major conformational difference is observed compared to the previous structure. The difference in ECL conformation can be induced by their ligand, e.g. its additional substituent opening up the extended binding pocket that they describe, or the Fab, or both, and it's unclear if this difference is important for drug design, i.e. makes their structure more useful for drug design than the previous structure. It is clear from the structure that Fab stabilizes the ECL conformation, and in part protrudes into the D2-binding site (e.g. residues Phe54 and Tyr55 of Fab chain C), pushing outward D2-TM5. The authors don't provide functional or binding data in presence of Fab, or indeed any characterization of Fab beyond the structure itself – a comparable study (EP4 structure by Toyoda et al 2018) much more carefully characterized the Fab used there. Without any characterization of Fab (binding and functional) the biological relevance of the changes the authors observe can't be estimated.

Fig. A. Gel Filtration

We appreciate these concerns. We were unable to measure the antagonist activity and the binding activity of the Fab3089-receptor complex of the wild-type and mutant D₂R_s. As shown in Fig. A, Fab3089 does not bind solubilized wild-type D₂R (Fig. A-b) or the solubilized ΔN-D₂R (N-terminal deletion D₂R) (Fig. A-c). We prepared ΔN-D₂R against the possibility that the N-terminal region inhibits Fab3089 from binding to D₂R. Moreover, Fab3089 binds the crystallized construct only after it is purified (Fig. A-a, d). These results suggest that components in the cell membrane inhibit the Fab3089 binding to D₂R. Indeed, when we measured the antagonist activity of the wild-type and mutant D₂R_s in the presence of Fab3089, the activities were the same as those without Fab3089, probably because Fab3089 does not bind to the receptor in the membrane. For purification, a ligand must be bound to D₂R to stabilize the receptor. Therefore, a measurement of the binding affinity is impossible because it requires a ligand-free receptor.

Can both D₂-spi-Fab and D₂-ris structures be correct, and observed differences be inherent to the difference of systems studied? – e.g. Roth lab describe no effect of mutating ECL2 residue 184 on binding/kinetics of several ligands, while the present study finds a large effect on receptor function in presence of their (different) ligand. The authors should test the 184A mutation in their binding assay also, not only functional,

and they should test effect of this mutation on the risperidone ligand using their assays to exclude assay artefacts.

The expression level of I184A was very low, and we failed to perform a binding assay for I184A. We measured the antagonist activity of I184A for risperidone (Supplementary Table 3) and found it similar to that of the wild-type ($\Delta pK_B = -0.44$), suggesting that I184 does not contact risperidone. Thus, we believe that both D_2R_{spi} and D_2R_{ris} are correct.

Perhaps the ECL2 is dynamic and does not interact with risperidone, but it interacts with spiperone – they comment that this is unlikely (line 327) based on comparison to structure of 5-HT_{2A}-ris complex structure, but I don't find such a comparison between two classes of receptors very convincing, even though they share ligands, absent corroborating experimental results.

We agree with the reviewer and have deleted the paragraph accordingly.

Reviewer #2 (Remarks to the Author):

Comments for Authors:

The presented manuscript 'Structure of the dopamine D2 receptor in complex with the antipsychotic drug spiperone' by So Iwata, Tatsuro Shimamura and colleagues characterizes the structural basis of the dopamine D2 receptor (D2R) in complex with an antipsychotic drug, the D2R-antagonist spiperone. The authors combine XFEL structure data of the D2R-spiperone complex (D_2R_{spi}) and several interesting substitutions of D2R binding pocket as well as a very comprehensive comparison with the available structures of D2R in complex with risperidone (D_2R_{ris}) and other Dopamine and Serotonin receptors D3R, D4R and 5-HT_{2A}(C)R. The D_2R_{spi} structure shows some decisive differences in the ligand-binding pocket compared with D_2R_{ris} . They also found some explanations for the high specificity of spiperone to the D2R and 5-HT_{2A}R. The structural data look well refined in relation to the limitation in the resolution of 3.2Å. Also all other included data are technically well done. The value of new information in this study is given by the new and extensive structural comparisons. The new structure information of the D_2R_{spi} structure is, however, somewhat limited and for my feeling not substantially more or less artificial than in the

already known crystal structure D2R_{ris} (Nature 2018). In addition, many mutants in D2R have already been investigated before.

The manuscript would benefit a lot from (a bit) shortening the text (e.g. combining structure comparisons) and making the images clearer, and above all not talking down about the other D2R_{ris} structure in every section.

Nevertheless, I think that this study and the new data can contribute to a better understanding and to new approaches for D2R-ligand development.

We wish to express our appreciation to the reviewer for these insightful comments, which have helped us significantly improve the paper. We have shortened the text in the structure comparison sections and made the images clearer. Furthermore, we have removed negative comments about D₂R_{ris}.

We provide a response to each of your specific comments below.

Remarks:

- (line 76-77). The authors could highlight/label the PIF motif in figure 2a. (line 154/156) Is there really a direct interaction between Ile122 of PIF and spiperone? What is the exact distance?

We have changed the color of the PIF motif in Fig. 2a to help it stand out. The distance between Ile122 and spiperone was 3.7 Å (page 9, line 172).

- (line 83-84): The sentence is misleading. Spiperone binds to several other receptors (D3R, D4R etc.)

We have changed the sentence to read as follows: “we describe the structure of D₂R in complex with spiperone (D₂R_{spi}), a butyrophenone typical antipsychotic that binds with high affinity to D₂R, D₃R, D₄R, and 5-HT_{2A}R.” (page 5, lines 76–77)

- The authors could improve Figure 1. The blue color in a and c and the depth of field of the receptor image are not so catchy (with a better positioning of the labels).

We have remade Fig. 1 to resolve the concerns raised by the reviewer.

- (line 77-81): The authors should explain this small paragraph in a better way and at more detail. Maybe a visualization with a new supplemental figure would help. For example, ECL1 has in D2R, D3R and D4R almost the identical conformation. Only Trp100 in D2R looks different and is directed towards the binding pocket. On the other hand, ECL2 is completely different in D2R compare with D3R and D4R. In both structures, D2R_{ris} and D2R_{spi} the residue Trp100 is important for the antagonist binding.

Thank you for this advice. We have changed the paragraph as suggested and added a new supplementary figure for clarity (Supplementary Fig. 1) (page 5, lines 67-74)

I didn't understand why the D2R_{ris} structure is her more artificial. It is a different antagonist and maybe induce a different binding mode. The argumentation here is too far-reaching and both structures are biased by their crystallization constructs.

We agree with the reviewer. In the revised manuscript, we treat both D₂R_{ris} and D₂R_{spi} as representing native conformations.

- (line 97-98): The authors should determine the affinity (K_d) of the wild-type D2R themselves (Table 1). The affinity of the published (in 1990) wild-type D2R to spiperone is different compared to both crystallization constructs.

We have determined the K_d value of the wild-type D₂R, as shown in Supplementary Table 2.

- The comparison of D2R_{ris} and D2R_{spi} shows a clear 2 Å motion of TM6. How exactly is this motion compared with D3R_{eti} and D4R_{nem}. The outward motion of the TM6 at the binding pocket also seems to be significantly different in D3R and D4R.

The distance shown in Fig. 3a is the longest distance between the C α atoms of the corresponding residues on TM6 in D₂R_{spi} and D₂R_{ris} superposed by the program SSM Superpose in COOT. The longest distance of TM6 is at its extracellular end. After refinement using reprocessed data, the movement is 2.2 Å. The movement of TM6 between D₂R_{spi} and D₃R_{eti}, and between D₂R_{spi} and D₄R_{nem} were 4.1 and 5.1 Å, respectively, at the extracellular end of TM6. However, the side-chain positions of Phe^{6.52} and His^{6.55} were similar between D₃R_{eti} and D₄R_{nem}.

- (line 125-): The contact to of Ile183 seems to be not important (SI Table2) which is absolute in line with the D2Rris structure (no contact). Maybe the ECL2 has here a different functional interaction in both cases, D2Rris vs. D2Rspi, (or as an intermediate conformation of ECL2 in D2Rris). For my feeling the variations of ECL2 per se is the interesting point here.

The antagonist activity of I183A is less affected for risperidone ($\Delta pK_B = 0.48$) than for spiperone ($\Delta pK_B = 0.79$), as shown in Supplementary Table 3. We suppose Ile183 does not contact risperidone.

- A very important difference is the different orientation of F110 in D2Rris vs. D2Rspi. In D2Rspi F110 is facing to the solvent, but this is clear due sterically hindrance of the phenyl ring from spiperone. In risperidone this phenyl ring does not exist, so the interaction between the two antagonists a D2R must be different at this point. That is in very interesting feature. What happens if this is mutated to Trp (as in 5-HT2a-R).

We thank the reviewer for the comment and agree that this is a very important difference. F110W decreased the antagonist activity for spiperone, as shown in Supplementary Table 3.

- (line 193-194): The sentence is very misleading. Ligand binding is not a static process.

We agree with the reviewer and have deleted these sentences.

- (line 207-210): The sentence clearly goes too far. As asked above, how is the contact distance in the D2Rspi structure from Ile122 to spiperone? Risperidone is also clearly a different ligand than spiperone. Both bind simply differently in some details (different head group like flourophanyl etc). And as a remark, as the authors have explained very nicely in their own publication (Kimura et al. NSMB 2019) about the 5-HT2AR: ‘5-HT2ARris, the entrance of the ligand-binding pocket between TM7 and ECL2 is wider by up to 2.2 Å than that in 5-HT2ARzot, which is essential for binding to the tetrahydropyridopyrimidinone ring of risperidone’. The different ligands deform the binding pocket also simply different.

We agree with the reviewer and have deleted the sentence in question. The distance between Ile122 and spiperone is 3.7 Å.

- (line 219-221): The sentence is misleading. The pockets are obviously different in more than one amino acid residue (F/W3.28). This is nicely shown in figure 4e. The authors could name other strong differences such as L2.64, which is really different from the other receptors, 5-HT_{2A}R_{zot} and 5-HT_{2A}R_{ris}.

In the original manuscript, we wanted to say that although the residues in the ligand-binding pocket are not entirely conserved between D₂R and 5-HT₂R_s, the C α atoms of these residues locate to a similar position between them. In the revised manuscript, however, we have deleted the first several sentences, including the sentence in question, for brevity. Indeed, most deleted sentences were already mentioned in the introduction section.

- (line 233-254): This is one potential explanation. The authors should put it that way. I am not so convinced because the receptors are very dynamic. The D₂R pairs are F and W and in the 5-HT structures W and L. They also could investigate a double mutation F130W3.28 and W90L2.60 for a comparison with the 5-HT structures. But it is surely one crucial site for ligand selectivity.

We have rewritten these sentences (page 12, lines 242–247). Furthermore, we determined the antagonist activity for spiperone in the W90L/F110W mutant. As shown in Supplementary Table 3, the mutant significantly decreased the activity ($\Delta pK_B = -1.43$).

- (line 266-268): This is also true for D₂R_{ris} (larger TM6 movement).

We have added D₂R_{ris} and D₂R_{hal} in the sentence (Page 12, line 263).

- (line 333-335) : To my mind this sentence in this form is really unnecessary. Both structures have more or less their justification for a structure-based drug design.

We agree the reviewer. We have deleted these sentences.

- Figures 3b and 4a,b are overloaded and really hard to distinguish. In Figure 4 the same color disturbs for 5-HT_{2A}R_{ris} vs D₂R_{spi}.

We have adjusted these figures for better readability.

- The sentence in figure 3 is partially not correct? The text reads as follows ‘...residues in D2Rspi and the ... in D2Rris are shown in cyan and magenta sticks, respectively. Red arrows indicate the shift of helices in D2Rris with distance relative to D2Rspi.’ It should revise to ‘Red arrows indicate the shift of helices in D2Rspi with distance relative to D2Rris.’

Thank you for catching this mistake. We have corrected the legend of Fig. 3 accordingly.

- The 2fo-*fc* electron density in supplemental figure 1c is not very nice. I see almost nothing but a blue blob? The authors should improve this figure significantly.

We have remade the figure. It is now Supplementary Fig. 4c in the revised manuscript.

- A second different view of supplemental figure 1d would be very insightful.

We have corrected the figure, as suggested. In the revised manuscript, the figure is now Supplementary Fig. 4d.

Reviewer #3 (Remarks to the Author):

This manuscript describes a new crystal structure of dopamine D2 receptor (D2R) in complex with the antagonist spiperone (D2Rspi). There is one previous D2R structure in complex with risperidone (D2Rris). This previous structure required several thermostabilizing mutations, which are not present in the current structure (however, two other mutations were introduced in D2Rspi, see below). The D2Rspi presented in this manuscript is different from the previous D2R structure in several striking ways in or near the ligand binding pocket, in particular in the conformations of ECL1, ECL2 and TM5. The author also compared D2Rspi to other related 5-HT_{2A}Rris, 5-HT_{2C}Rrit, D3Reti and D4Rnem structures. They identified the structural basis for the high-affinity of spiperone binding for D2R and 5-HT_{2A}R but not for 5-HT_{2C}R, i.e., the residue configurations at positions 3.28 and 2.60. They conclude that the D2R, D3R, and D4R conformations may adapt to different types of antipsychotics. The authors also carried out systematic mutagenesis study to validate the spiperone binding residues identified in the D2Rspi structure.

This new D2Rspi structure together with previous D2Rris, D3Reti and D4Rnem provide important clues for the field to understand the ligand-receptor recognitions for these highly-homologous and therapeutically important receptors. I expect the manuscript to be of high impact because unlike many new structures that reveal only incremental advances, this structure suggests that different antagonists stabilize quite different receptor conformations, which will have major impact not only on virtual screening for new ligands but also on potential insights into biased signaling. While these ideas are only hinted at in the manuscript, the work is well supported by the mutagenesis data, making the results an important contribution to the literature.

We thank the reviewer for the time taken in carefully reviewing our manuscript and for the appreciation of our manuscript and its potential impact. We have answered each of your comments below.

I only have a few suggestions,

For the statement in the abstract, “D2Rris exhibits artificial conformation in the ligand-binding pocket owing to a mutation introduced for the stability, limiting efficient development of antipsychotics”, the authors need to elaborate why the conformation is artificial. Note that the impact of I122A on binding is relatively small, and the risperidone pose is similar in D2Rris and 5-HT2ARris (see supplementary figure 4), which does not have a mutation at the aligned position (ref 36). I think it remains an open question whether the mutation led to an artificial pose or whether the different drug scaffolds can stabilize different conformations, and this should be discussed. The authors may also want to provide data to support why they believe that D2Rris may limit efficient development of antipsychotics.

We agree with the reviewer’s comment that the impact of I122A on binding in D₂R_{ris} is relatively small and that it remains an open question whether the mutation leads to an artificial pose or whether the different drug scaffolds can stabilize different conformations. Moreover, D₂R_{hal} was used for the structure-based discovery of selective ligands. Therefore, we have omitted the statement containing “artificial” from the revised manuscript.

A related issue is that the current structure also has thermostabilizing mutations, albeit different. Thus, this structure is subject to the same consideration. Is the structure “artificial” or does it represent a different conformation with a different scaffold that

naturally occurs, aided by the mutations. This is likely unknowable but it seems inappropriate to criticize the previous structure without applying the same caveat here, i.e., the impact of the S121K and L123W mutations introduced in the D2Rspi structure. Curiously, the Kd in supplementary table 1 are in nM ranges, while the spiperone affinity has been more often reported in pM range (see literature and compare supplementary tables 1 and 2). If “D2R-bRIL”, a NT-truncated and IL3-replaced construct, which is not exactly the background construct of “D2R-mbIIG S121K3.39/L123W3.41” with a different IL3 replacement, has a disrupted spiperone binding already, it is critical to understanding how the mutations impaired binding themselves. Interestingly, an examination of the structure reveals that the sidechain of S121K protrudes in the Na⁺ binding site. As Na⁺ binding may potentially affect the binding of particular ligands as well as the receptor conformation (see Neve et al., Mol Pharm 1991, 2001; Michino et al., Chem Commun 2015), this may be relevant to the arguments in this manuscript. The impact of the S121K and/or L123W need to be tested and compared with wt to supplement the results shown in supplementary table 2. Alternatively, if these two mutants render the receptor functionally inactive, this should be reported and radioligand binding studies should be performed to compare the affinity of spiperone at this mutant and the WT receptor. It would not be unreasonable to also add functional inhibition or binding affinity data on a Na⁺ dependent ligand, such as eticlopride or sulpiride to ascertain the impact of the potential disruption of the Na⁺ site. These data seem important to characterizing the construct used to generate this interesting structure.

We measured the antagonist activity for spiperone of S121K, L123W, and S121K/L123W. L123W showed similar activity with the wild-type, as shown in Supplementary Table 3. The antagonist activity of S121K and S121K/L123W could not be determined because the S121K replacement stabilizes the inactive conformation and prevents activation by dopamine (Supplementary Fig. 3). However, we measured the binding affinity for spiperone of the wild-type, S121K, L123W, S121K/L123W, and the crystallized constructs, as shown in Supplementary Table 2. They all show a similar affinity for spiperone, suggesting the mutations do not affect the spiperone binding.

We noticed that ligand depletion occurred in the ligand-binding assay for spiperone. Therefore, we changed the assay method by increasing the assay volume and using the equation accounting for ligand depletion in GraphPad Prism 5 software. Thus, the Kd

value of the crystallized construct in the revised manuscript ($K_d = 0.29$ nM) is different from that in the first manuscript ($K_d = 3.0$ nM).

We also measured the binding affinity of S121K for eticlopride. The S121K mutation decreased its affinity to eticlopride (Supplementary Table 2), suggesting that the side chain of Lys121^{3,39} does not sufficiently mimic the sodium ion in the allosteric binding site for the binding of eticlopride.

On page 18, the first paragraph of discussion, the I183C results from ref 23 seem to be slightly over-interpreted. I183C was shown to have low accessibility that was only detected by MTSET, so it is less likely to be in direct contact with ligand but is more likely directed toward the ligand-binding pocket.

We have corrected the paragraph accordingly (page 14, lines 292–293).

Interestingly, in Supplementary Table 2 of the current manuscript, I183A appears to improve spiperone's affinity by 6 fold. Given that the sidechain of I183 in the crystal structure appears to have been stabilized by a Tyr from Fab3089, the authors need to be more cautious in interpreting the interaction between I183 and spiperone.

We have mentioned in the text that contact between Ile183 and spiperone can be influenced by the binding of Fab3089, because the side-chain conformation of Ile183^{45,51} is stabilized by the Tyr55 of Fab3089 (Fig. 2c). The I183A mutant slightly increased the antagonist activity for spiperone (Supplementary Table 3) (pages 7-8, lines 140–143).

REVIEWERS' COMMENTS

Reviewer #1 (Remarks to the Author):

The changes to the revised manuscript are to my satisfaction, however I believe the authors should generate a "traditional" simulated annealing composite omit map around the ligand, and present it at least as prominently as their polder map, so that readers less familiar with the latter have a point of reference for the ligand density.

REVIEWERS' COMMENTS

Reviewer #1 (Remarks to the Author):

The changes to the revised manuscript are to my satisfaction, however I believe the authors should generate a "traditional" simulated annealing composite omit map around the ligand, and present it at least as prominently as their polder map, so that readers less familiar with the latter have a point of reference for the ligand density.

Response to this comment.

We thank the reviewer for the positive comment and careful review. We agree with the reviewer and have added a simulated annealing composite omit map around the ligand in Supplementary Fig. 4d.